# Coronavirus endoribonuclease nsp15 suppresses host protein synthesis and evades PKR-eIF2α-mediated translation shutoff to ensure viral protein synthesis

**Xiaoqian Gong[1,2☯], Shanhuan Feng[1☯], Jiehuang, Wang[1☯], Bo Gao[1], Wenxiang Xue[1], Hongyan Chu[1], Shouguo Fang[3], Yanmei Yuan[1], Yuqiang Cheng[4], Min Liao[5], Yingjie Sun[1], Lei Tan[1], Cuiping Song[1], Xusheng Qiu[1], Chan Ding[1,6], Edwin Tijhaar[2], Maria Forlenza[2,7]\*, Ying Liao[1]\***

**1** Department of Avian Infectious Diseases, Shanghai Veterinary Research Institute, Chinese Academy of Agricultural Sciences, Shanghai, China, **2** Cell Biology and Immunology Group, Wageningen University and Research, Department of Animal Sciences, Wageningen, the Netherlands, **3** College of Agriculture, College of Animal Sciences, Yangtze University, Jingzhou, China, **4** Shanghai Key Laboratory of Veterinary Biotechnology, Key Laboratory of Urban Agriculture (South), School of Agriculture and Biology, Shanghai Jiao Tong University, Shanghai, China, **5** Key Laboratory of Animal Virology of Ministry of Agriculture, Zhejiang University, Hangzhou, China, **6** Jiangsu Co-innovation Center for Prevention and Control of Important Animal Infectious Diseases and Zoonoses, Yangzhou University, Yangzhou, China, **7** Host-Microbe Interactomics Group, Wageningen University and Research, Department of Animal Sciences, Wageningen, the Netherlands,

☯: Contributed to the work equally as the first author
\* liaoying@shvri.ac.cn (YL); maria.forlenza@wur.nl (MF)

## Abstract

The endoribonuclease (EndoU) nsp15 of coronaviruses plays a crucial role in evading host innate immune responses by reducing the abundance of viral double-stranded RNA (dsRNA). However, our understanding of its interactions with host cellular targets remains limited. In this study, we demonstrate that overexpression of nsp15 from four coronavirus genera inhibits cellular protein synthesis and causes nuclear retention of PABPC1. Mutation analysis confirms the essential role of EndoU activity in these processes. Fluorescence in situ hybridization (FISH) analysis shows that cellular mRNA co-localizes with nsp15 in certain cells. Real time RT-PCR indicates that the mRNA levels of several antiviral genes decrease in cells expressing nsp15, and this reduction depends on the EndoU activity of nsp15. Using infectious bronchitis virus (IBV) as a model, we investigate the inhibitory effect of nsp15 on protein translation during infection. We find that infection with IBV with functional nsp15 suppresses protein synthesis in a PKR-eIF2α independent manner, with PABPC1 mainly located in the cytoplasm. However, infection with EndoU activity-deficiency mutant virus rIBV-nsp15-H238A results in the accumulation of viral dsRNA, triggering a PKR-eIF2α-dependent shutdown of protein synthesis and leading to the nuclear relocation of PABPC1. In the absence of the PKR-eIF2α pathway, IBV is still able to suppress host protein synthesis, while the inhibitory effect of rIBV-nsp15-H238A on protein synthesis was significantly reduced. Although nsp15 locates to replication-transcription complex (RTC) during infection, RNA immunoprecipitation

**Data availability statement:** All relevant data are within the manuscript and its Supporting Information files. The raw data can be found at the following link. https://zenodo.org/records/14580340?preview=1&token=eyJhbGciOiJIUzUxMiJ9.eyJpZCI6IjVmN-jEyZDEwLWVmMjEtNGJhMS05NmI2LTk2OTZIMGEwZWViOCIsImRhdGEiOnt9LCJyYW5kb20iOiIwYWYxNTYxNmFlYTg3YmEzOGE1MjNkYWI0OMzJiNmI3YSJ9.u88n2K8v4Fq7ezOuYjf5-szf2nwD_NQBn2JpjrdSl41DvmxhNUT6Qixz_HeZ3mF9zIenCOYx8WC-2yGGYdiOyg

**Funding:** This work was supported by the National Natural Science Foundation of China (32172834 and 32372999) awarded to Y.L., the National Key Research and Development Program (2021YFD1801104) awarded to Y.L., and the Shanghai Natural Science Foundation (23ZR1477000) awarded to Y.L. The funders had no role in study design, data collection and analysis, decision to publish, or preparation of the manuscript.

**Competing interests:** The authors have declared that no competing interests exist.

(RIP)-Seq analysis confirms that IBV nsp15 binds to six viral RNAs and 237 cellular RNAs. The proteins encoded by the nsp15-associated cellular RNAs predominantly involved in translation. Additionally, proteomic analysis of the nsp15 interactome identifies 809 cellular proteins, which are significantly enriched in pathways related to ribosome biogenesis, RNA processing, and translation. Therefore, nsp15 helps virus circumvent the detrimental PKR-eIF2α pathway by reducing viral dsRNA accumulation and suppresses host protein synthesis by targeting host RNAs and proteins. This study reveals unique yet conserved mechanisms of protein synthesis shutdown by catalytically active nsp15 EndoU, shedding light on how coronaviruses regulate host protein expression.

## Author summary

Coronavirus infection is known to suppress host protein synthesis, primarily through the actions of α- and β-coronavirus nsp1. However, the mechanisms by which γ- and δ-coronaviruses inhibit host protein translation, despite lacking nsp1, remain unclear. Our study reveals that the coronavirus EndoU nsp15 plays a crucial role in inhibiting host protein translation, by targeting cellular RNAs and proteins related to RNA processing, translation, and ribosome biogenesis. In the other way, by reducing the levels of viral dsRNA, IBV nsp15 helps the virus evade the host's antiviral response, including PKR-eIF2α-dependent translation shutoff that could otherwise affect both viral and host mRNA translation. This novel finding elucidates how nsp15 aids coronaviruses in evading the host innate immune response and facilitating virus replication.

## Introduction

Coronaviruses are a diverse group of viruses classified into four genera: α, β, γ, and δ. They are enveloped, positive-sense, single-stranded RNA viruses that possess the largest known RNA genome, ranging from approximately 25 to 32 kilobases [1]. Upon entry into host cells, the coronavirus genome serves as a template for the synthesis of polyproteins 1a and 1ab, which are encoded by two large open reading frames (ORF1a and ORF1b) occupying over two-thirds of the genome at the 5'-end. Translation of ORF1b requires a programmed -1 ribosomal frameshifting mechanism to produce polyprotein 1ab [2]. These polyproteins are cleaved by internal papain-like protease (nsp3) and 3C-like protease (nsp5) to produce non-structural proteins (nsps) [3]. The α- and β-coronaviruses encode 16 nsps (nsp1 to nsp16) [4]; while the γ- and δ-coronaviruses encode 15 nsps (nsp2-nsp16) and lack the most N-terminal cleavage product nsp1 [5,6]. Several nsps contain domains crucial for the replication and transcription of viral RNA. These include the RNA-dependent RNA polymerase (RdRp) nsp12, the primase nsp7 and nsp8, the RNA helicase/5'-triphosphatase nsp13, the exoribonuclease nsp14, the EndoU nsp15, and the RNA-cap methyltransferase nsp14 and nsp16 [7–13]. Among these, RdRp nsp12 plays a central role in orchestrating the replication and transcription of viral RNA, while other nsps serve supportive functions. These diverse nsps collectively contribute to various aspects of the coronavirus lifecycle, including replication, transcription, and evasion of host immune responses. The highly conserved EndoU nsp15 has been reported to be an integral component of the replication and transcription complex (RTC) in various coronaviruses [14–16]. It possesses uridylate-specific endonucleolytic activity on viral RNA within the RTC. During infection with coronaviruses like SARS-CoV-2, SARS-CoV, MERS-CoV,

MHV, and IBV, the RTC is localized within virus-induced double membrane vesicles (DMVs) [17–21] or zippered endoplasmic reticulum (ER) and spherules single membranes [22]. These structures are formed by virus encoded transmembrane proteins such as nsp3, nsp4, and nsp6 [23,24].

To achieve successful replication, coronaviruses utilize various strategies to evade or counteract host anti-viral response [25]. The viral ligand dsRNA, an intermediate product of viral replication, is founded within the remodelled intracellular membrane structures, representing as the products of ongoing viral RNA replication [19,26]. Nsp15 activity plays a crucial role in controlling the levels of dsRNA to evade detection by host cell sensors, thereby enabling the virus to evade the host innate immune response. This function of nsp15 has been reported in various coronaviruses, including MHV, HCoV-229E, PEDV, and IBV [27–30], and is likely also applicable to SARS-CoV-2 [31,32]. Although the role of nsp15 in cleaving viral RNA and facilitating innate immune escape is well established, its interaction with host cells remains poorly understood. Our previous study has indicated that nsp15 suppresses the formation of stress granules, which are cytoplasmic aggregates of RNA and proteins induced under conditions of cellular stress including virus infection [30]. Moreover, nsp15 promotes the nuclear accumulation of the cytoplasmic poly(A) binding protein (PABPC1) [30], implying that nsp15 may not only target viral RNA but also influence host cell functions by affecting unknown host substrates. Further investigation is necessary to comprehensively elucidate the breadth of nsp15's interactions with both viral and host components, as well as its implications for coronavirus replication.

Inhibiting host gene expression is both an alternative strategy to evade the host innate immune response by reducing antiviral protein synthesis and a means to hijack the host translation machinery to support viral mRNA translation. Eukaryotic gene expression encompasses several processes, including transcription, RNA processing, nuclear RNA export, protein synthesis, and post-translational modification [33]. Viruses can suppress host gene expression through various mechanisms, such as reducing the levels of host mRNA or interfering with their association with ribosomes or translation initiation factors. For example, poliovirus 3C protease inhibits RNA polymerase II mediated transcription initiation by cleaving transcription activator Oct-1 [34,35]; influenza A virus (IAV) polymerase acidic (PA) protein snatches the capped primers from nascent host transcripts to synthesize viral mRNA [36]; IAV NS1 protein inhibits polyadenylation of cellular precursor mRNA (pre-mRNA) and blocks the nuclear export of cellular mRNA [37–39]; human immunodeficiency virus (HIV) viral protein R (VPR) inhibits the splicing of host pre-mRNA [40]; herpes simplex virus (HSV) infected cell protein 27 (ICP27) hinders host transcription termination [41]; poliovirus 2A protease mediates nucleoporin cleavage, affecting cellular mRNA nuclear export [42]; poliovirus 2A protease also cleaves the initiation factor eIF4G and this cleavage is mainly responsible for the inhibition of cap-dependent translation of host mRNAs [43,44]; poliovirus 3C proteinase cleaves PABP to inhibit translation initiation [45]. It has been reported that several $\alpha$- and $\beta$-coronaviruses utilize their nsp1 to manipulate the host translation machinery through various mechanisms, including: repressing host mRNA transcription in the nucleus [46], preventing the nuclear export of host mRNA [47,48], degrading host mRNA in both the nucleus and cytoplasm [46,49–52], interfering with ribosomes to inhibit host mRNA translation [51,53,54]. However, for $\gamma$-coronavirus and $\delta$-coronavirus, which lack nsp1, although host protein synthesis shutoff is observed, the precise mechanisms and proteins involved remain poorly characterized [55–57]. Notably, in IBV, the 5b and S protein have been reported to be involved in translation inhibition [55,58].

Several viral endonucleases are involved in regulating not only viral RNA but also host gene expression by targeting host mRNA for degradation. Examples include: HSV virion host

shutoff (VHS) protein [59–61], Kaposi's sarcoma-associated herpesvirus (KSHV) shutoff and exonuclease (SOX) protein [62–64], Epstein Barr virus (EBV) BGLF5 protein [65–67], and murine herpesvirus 68 (MHV-68) muSOX protein [62,68]. These viral proteins possess endonuclease activity and inhibit synthesis of cellular proteins by promoting the degradation of cellular mRNAs [62,64,69,70]. The cap-dependent endonuclease PA of IAV is associated with the RNA polymerase complex and responsible for snatching capped oligonucleotides from cellular pre-mRNAs and using them as primers for the synthesis of viral mRNAs [36,71,72]. This process aids viral gene expression while suppressing host mRNA maturation. Additionally, IAV encodes another endonuclease, PA-X, that selectively degrades host RNAs and usurps the host mRNA processing machinery to destroy nascent mRNAs, thereby limiting host gene expression [73–75].

Here we investigated the ability of EndoU nsp15 encoded by coronaviruses to inhibit cellular protein synthesis. Our findings provide evidence that nsp15 inhibits host protein synthesis by targeting host factors involved in the translation complex, including host RNA and PABPC1. This mechanism requires the EndoU activity of nsp15. In the context of IBV infection, nsp15, likely in coordination with other IBV proteins, triggers an eIF2α-independent host translation shutoff. However, when nsp15's EndoU activity is deficient, increased viral dsRNA accumulates in cells, leading to protein synthesis inhibition through a PKR-eIF2α-dependent mechanism, which hinders both host and viral mRNA translation initiation. Thus, nsp15 helps evade the virus-detrimental PKR-eIF2α-mediated shutdown of protein translation by reducing viral RNA accumulation. RIP-Seq RNA interactome and proteomic interactome analyses demonstrate that nsp15 not only binds with viral RNAs, but also interacts with cellular RNAs and proteins, which are enriched in pathways related to RNA processing, translation, and ribosome biogenesis. Overall, our findings reveal new yet conserved strategies shared among coronaviruses for suppressing host gene expression, highlighting the crucial role of EndoU nsp15 in these processes.

## Results

### IBV nsp15 suppresses exogenous protein synthesis

We previously demonstrated that IBV nsp15 disrupts stress granules formation, likely by targeting the host translation machinery [30]. Furthermore, during our screening of IBV-encoded type I interferon antagonists using a luciferase reporter system, we observed that IBV nsp15 not only reduced the expression of luciferase driven by the IFNβ promoter but also attenuated the expression of co-transfected plasmids encoding HA-tagged MAVS (HA-huMAVS) in 293T cells (S1 Fig). These observations led us to hypothesize that IBV nsp15 interferes the host translation. Simultaneously, we found that proteins 5a and E similarly suppressed the co-transfected huMAVS expression and concurrently decreased the expression of IFNβ-driven luciferase. This suggests that IBV encodes multiple proteins involved in inhibiting protein expression.

In this study, we focus on nsp15 to confirm whether it indeed helps the virus inhibit host protein synthesis and to elucidate its mechanism of action. The IBV host cell line chicken embryo fibroblast DF-1 were co-transfected with plasmid encoding Flag-tagged IBV nsp15, along with various reporter plasmids, including V5-tagged constitutively active form of chicken MDA5 [V5-chMDA5(N)} [76], HA-tagged chicken MAVS (HA-chMAVS) [77], V5-tagged chicken interferon regulatory factor 7 (V5-chIRF7) [78], and enhanced green fluorescent protein (EGFP). As controls, cells were co-transfected with either vector PXJ40 or plasmids encoding Flag-tagged IBV nsp7, nsp8, nsp9, or nsp13, along with reporter plasmids. Western blot analysis showed that the overexpression of IBV nsp15 led to a significant

reduction in the protein levels of V5-chMDA5(N), HA-chMAVS, V5-chIRF7, and EGFP to varying degrees. In contrast, the overexpression of IBV nsp7, nsp8, nsp9, nsp12, or transfection of PXJ40 did not induce significant alterations in protein expression levels encoded by co-transfected reporter plasmids (Fig 1A). These findings conclusively demonstrate that nsp15 indeed suppresses the expression of exogenous proteins.

Since the suppression of exogenous protein expression by IBV nsp15 was not restricted to proteins involved in antiviral responses but was also observed for EGFP, we sought to investigate whether this effect on protein synthesis applies to host endogenous proteins as well. To explore this, we evaluated the signal of puromycin-labelled, newly synthesized endogenous peptides in cells expressing IBV nsp15. Puromycin, an analogue of tRNA, binds to elongating peptide chains, causing their premature termination [79]. Therefore, the signal from puromycin-labelled peptides, detected using an anti-puromycin antibody, provides a direct measure of the number of peptides synthesized *de novo* during the puromycin treatment period. DF-1 cells were transfected with either vector PXJ40 or plasmid encoding Flag-tagged IBV nsp15 and then subjected to indirect immunofluorescence. We observed a significant decrease in the puromycin labelling signal in all cells expressing IBV nsp15, while this effect was not seen in cells lacking nsp15 expression or those transfected with the vector PXJ40F (Fig 1B). These results collectively indicate that the overexpression of nsp15 inhibits the overall synthesis of endogenous proteins.

**The catalytic activity and oligomeric structure of IBV nsp15 are indispensable for inhibiting *de novo* protein synthesis**

The conserved catalytic residues essential for IBV nsp15 EndoU activity are located at histidine residues H223 and H238, while the residues critical for oligomerization are located at aspartic acid residues D285 and D315. In a previous study, we demonstrated that the EndoU activity of nsp15 is crucial for limiting the accumulation of viral dsRNA intermediates within IBV infected cells, thereby facilitating evasion of host recognition and delaying IFNβ production [30]. To ascertain whether the observed effect on *de novo* protein synthesis inhibition also relies on EndoU activity, we utilized previously generated mutated forms of nsp15 in which either one of the catalytic histidine residues or one of the aspartic acid residues was substituted with an alanine residue (H223A, H238A, D285A, D315A) [30]. Western blot analysis showed that mutating either the catalytic or the oligomerization core residues substantially abolished nsp15's inhibitory effect on the expression of exogenous proteins [chMDA5(N), chMAVS, chIRF7, EGFP] (Fig 2A), indicating the indispensable role of both EndoU activity and oligomerization in mediating inhibition of protein synthesis. Furthermore, it was observed that the expression level of wild-type nsp15 was lower compared to the levels of nsp15 mutants H223A, H238A, D285A, or D315A (Fig 2A), indicating that the impact on protein synthesis machine also influenced the expression of nsp15 itself. These findings validate that nsp15's inhibitory effect on cellular protein synthesis relies on both its EndoU activity and oligomeric structure. Interestingly, the expression levels of the oligomerization-deficient nsp15 mutants D285A and D315A were lower than those of the catalytic-deficient nsp15 mutants H223A and H238A, suggesting a potential impact on protein stability due to the inability to oligomerize.

Subsequently, we investigated the impact of wild-type and mutated nsp15 on endogenous *de novo* protein synthesis by analysing the fluorescence intensity of puromycin-labelled peptide signals. Indirect immunofluorescence analysis conducted in DF-1 cells revealed that cells expressing wild-type nsp15 exhibited a significantly reduced puromycin signal compared to cells not expressing nsp15. Conversely, this puromycin signal reduction was not observed in cells expressing mutated nsp15 H223A, H238A, D285A, or D315A (Fig 2B), highlighting the indispensable role of both EndoU activity and oligomerization structure in mediating the inhibitory effect on cellular protein synthesis. Furthermore, we evaluated the inhibitory

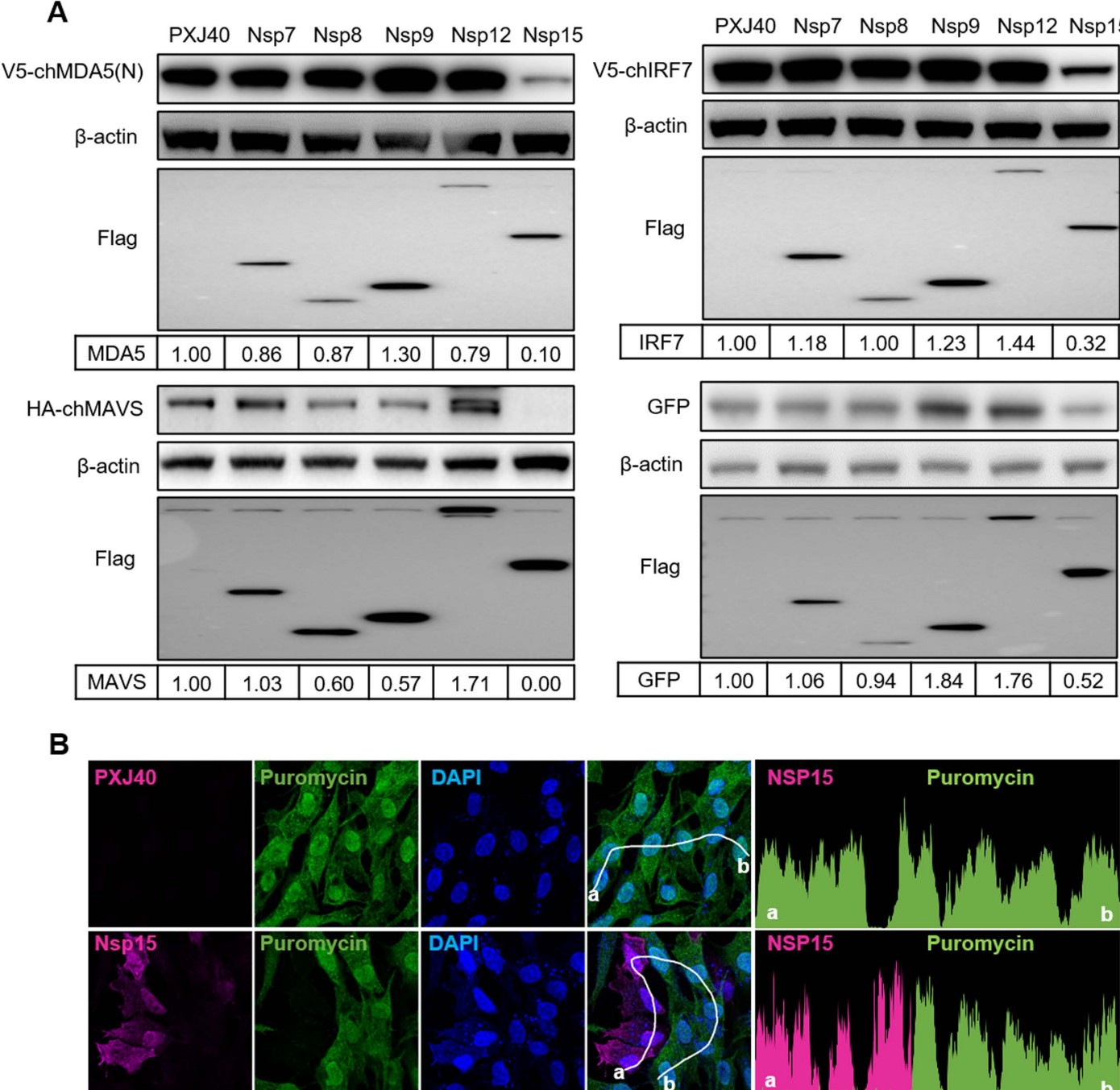

**Fig 1. IBV nsp15 inhibits *de novo* protein synthesis.** (A) DF-1 cells were co-transfected with plasmids encoding Flag-tagged IBV nsp7, nsp8, nsp9, nsp12, or nsp15 along with reporter plasmids encoding V5-chMDA5(N), HA-chMAVS, V5-chIRF7, or EGFP. The PXJ40 vector was included as a control. After 24 h, cells were collected for western blot analysis. Protein signals were detected using the indicated antibodies, with β-actin serving as a loading control. The density of the corresponding protein bands was analysed using ImageJ, normalized to β-actin, and the ratio was presented relative to PXJ40 transfected group. (B) DF-1 cells were transfected with a plasmid encoding Flag-nsp15 or PXJ40 for 23 h and treated with puromycin (5 μg/mL) for 1 h to label *de novo* synthesized peptides. Indirect immunofluorescence staining was performed to detect nsp15 (magenta) and puromycin-labelled *de novo* synthesized peptides (green). The nuclei were labelled with DAPI (blue). The fluorescence intensity of nsp15 and puromycin in individual cells along the white line (from a to b) is shown in the right panel (histogram plot). Representative images are shown.

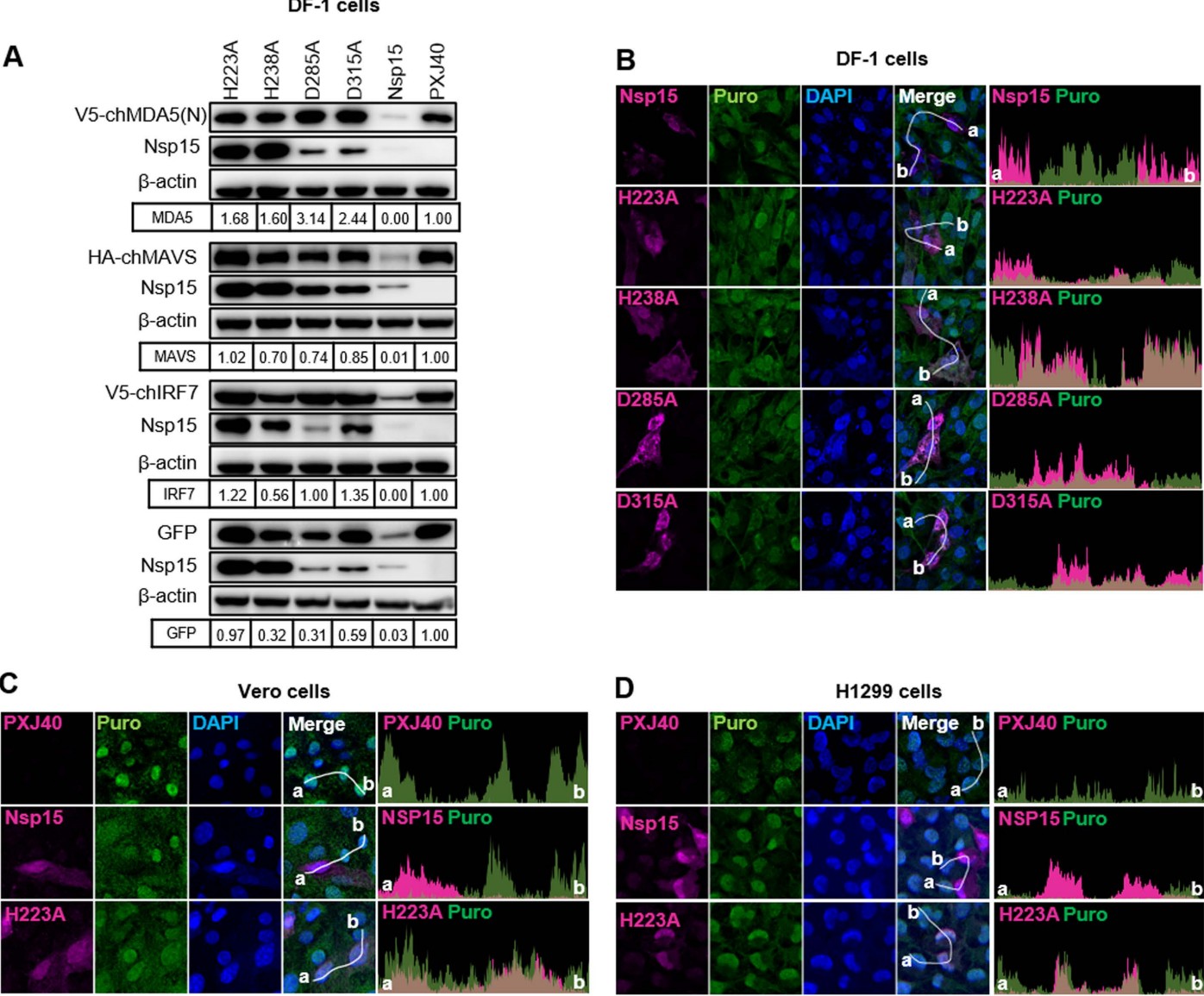

**Fig 2. The catalytic activity and oligomeric structure of IBV nsp15 are indispensable for the inhibition of *de novo* protein synthesis in chicken cells and mammalian cells.** (A) DF-1 cells were co-transfected with plasmids encoding Flag-tagged nsp15, catalytic-deficient nsp15 mutants (H223A, H238A), oligomerization-deficient nsp15 mutants (D285A, D315A), or the vector PXJ40, along with plasmids encoding V5-chMDA5(N), HA-chMAVS, V5-IRF7, or EGFP. After 24 h, Western blot analysis was performed using specific antibodies. β-actin served as a loading control. The density of the protein bands was analyzed using Image J, normalized to the signal of β-actin, and presented relative to the density detected in the corresponding PXJ40-transfected cells. (B-D) DF-1, Vero, and H1299 cells were transfected with plasmids encoding wild-type or mutated nsp15 and treated with puromycin (5 μg/mL) for 1 h at 23 h post-transfection (h.p.t). Indirect immunofluorescence staining was performed to detect nsp15 (magenta) and puromycin-labelled *de novo* synthesized peptides (green). Nuclei were stained with DAPI (blue). The fluorescence intensity of nsp15 and puromycin signal along the white line (from point a to b) is indicated by histogram plot. Representative images are shown.

effect of nsp15 on cellular protein synthesis in Vero and H1299 cells, which are permissive cell lines for the IBV-Beaudette strain. Consistent with the findings in DF-1 cells, indirect immunofluorescence analysis demonstrated that the presence of wild-type nsp15, but not the catalytic-deficient H223A, resulted in a lower puromycin labelling signal in Vero and H1299 cells (Fig 2C). Collectively, these results establish that the inhibitory effect of IBV nsp15 on

both exogenous and endogenous protein synthesis is a generalized phenomenon across different cell types, rather than being restricted to specific cell types. Both the catalytic activity and oligomeric structure are indispensable for nsp15 to inhibit *de novo* protein synthesis.

**Inhibition of *de novo* protein synthesis is a conserved feature of nsp15 from different genera of coronaviruses**

The conserved activity of catalytic histidine residues of nsp15 on inhibiting stress granule formation, as previously reported for nsp15 from IBV, PEDV, TGEV, PDCoV, SARS-CoV-1, MERS-CoV, and SARS-CoV-2 [30], prompts us to investigate whether the observed inhibitory effect of IBV nsp15 on protein synthesis is conserved among nsp15 from different genera of coronaviruses and whether such function is dependent on the EndoU catalytic activity. To address this, we co-transfected plasmids encoding wild-type or catalytic-deficient nsp15 from the aforementioned coronaviruses together with reporter plasmids encoding EGFP or IBV N into Vero cells. Western blot analysis revealed that wild-type nsp15 of PEDV, TGEV, PDCoV, and SARS-CoV-1 reduced the expression of both EGFP and IBV N, while catalytic-deficient nsp15 did not (Fig 3A-D). Conversely, nsp15 of MERS-CoV and SARS-CoV-2 did not show a pronounced effect on the expression of EGFP or IBV N (Fig 3E-F). These data suggest that, consistent with the finding for IBV nsp15, nsp15 from most of the tested coronaviruses can suppress the expression of exogenous proteins.

Next, we investigated whether nsp15 from different coronaviruses also suppresses endogenous protein expression. The nsp15 of porcine coronaviruses (PEDV, TGEV, PDCoV) was expressed in the porcine kidney cell PK15, while the nsp15 from human coronaviruses (SARS-CoV-1, MERS-CoV, SARS-CoV-2) was expressed in the human cell line HeLa. Consistent with the results for IBV nsp15, indirect immunofluorescence analysis revealed that PK15 cells expressing wild-type nsp15 of PEDV, TGEV, PDCoV (Fig 4A) and HeLa cells expressing wild-type nsp15 of SARS-CoV-1 displayed weaker puromycin labelling signals than cells not expressing nsp15 (Fig 4B). This effect was largely abolished in cells expressing catalytic-deficient nsp15. Interestingly, cells expressing nsp15 from MERS-CoV and SARS-CoV-2 exhibited only a moderate reduction in puromycin signal compared to cells not expressing nsp15. This reduction was less pronounced than that observed with nsp15 from other coronaviruses (Fig 4B), consistent with the findings presented in Fig 3E-F. This suggests that nsp15 from MERS-CoV and SARS-CoV-2 may function somewhat differently from nsp15 from other coronaviruses. Similar effects of nsp15 on puromycin staining were observed when nsp15 from PEDV, TGEV, and PDCoV was expressed in the porcine cell line PK1, ST, and PK15, respectively, as well as when nsp15 from SARS-CoV-1, MERS-CoV, and SARS-CoV-2 was expressed in Vero cells (S2 Fig). These findings suggest that the inhibition of protein synthesis by nsp15 EndoU is a general phenomenon in pan-coronaviruses and is not restricted to specific cell types.

**Nsp15 from various coronavirus genera promotes nuclear accumulation of PABPC1 and binds to cellular mRNA**

Previous findings from our study demonstrated that during stress conditions, nsp15 exhibits a conserved ability to induce the translocation of PABPC1 into the nucleus [30]. It has been shown that PABPC1 nuclear relocation is correlated with a reduction in cytoplasmic mRNA and regulates the nuclear-cytoplasmic trafficking of mRNA [80–82]. Considering that nsp15 suppresses *de novo* protein synthesis and its EndoU activity is essential, we next investigated whether nsp15 influences cellular mRNA abundance and its correlation with the nuclear relocation of PABPC1. We employed a cross-reactive PABPC1 antibody to facilitate

localization of PABPC1 did not result in obvious nuclear retention of cellular mRNA (red the visualization of PABPC1 localization in both DF-1 cells and mammalian cells. This was conducted concurrently with the examination of mRNA distribution through FISH, utilizing

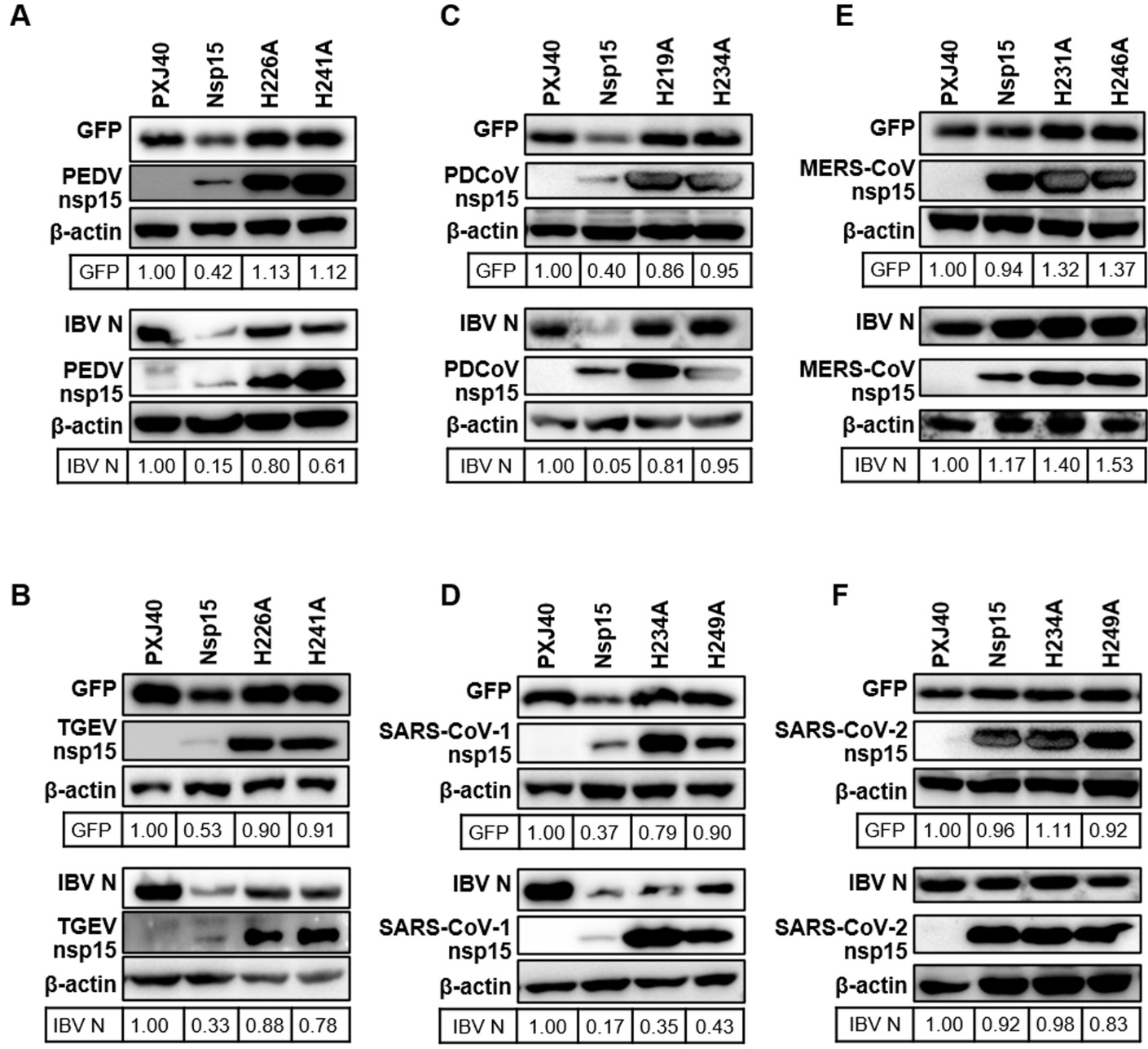

**Fig 3. Inhibition of exogenous protein expression by nsp15 from different coronaviruses.** Plasmids encoding wild-type or catalytic-deficient nsp15 from various coronaviruses were co-transfected with plasmids encoding EGFP or IBV N into Vero cells. After 24 h, Western blot analysis was conducted using specific antibodies. β-actin served as a loading control. Band densities of EGFP or IBV N were quantified using ImageJ, normalized to the signal of β-actin, and presented relative to the PXJ40 group.

fluorescently labelled oligo (dT) probes. In line with our prior finding, we found that the overexpression of IBV nsp15 induced nuclear localization of PABPC1 (yellow arrows) in DF-1 cells, a phenomenon not observed in cells expressing catalytic-deficient (H223A and H238A) or oligomerization-deficient (D285A and D315A) IBV nsp15 (Fig 5A). Notably, the nuclear arrows) (Fig 5B-D), unlike cells treated with sodium arsenite (ARS), where both PABPC1 and mRNA were restricted to the nucleus (Fig 5B). It was noted that the nsp15-induced nuclear

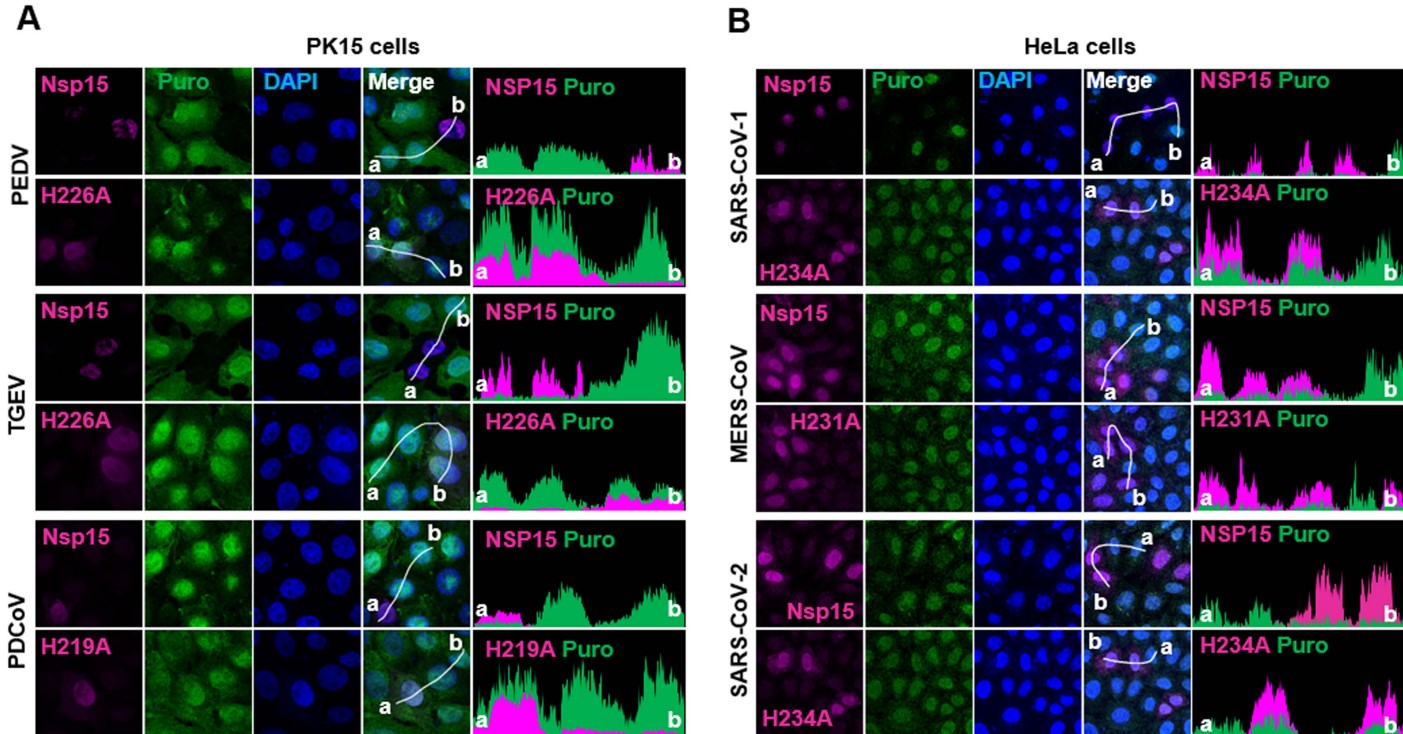

**Fig 4. Nsp15 from various coronaviruses inhibits *de novo* protein synthesis in PK15 cells and HeLa cells.** (A) PK15 cells were transfected with wild-type nsp15 or the corresponding catalytic-deficient nsp15 from porcine coronaviruses PEDV, TGEV, or PDCoV. (B) HeLa cells were transfected with wild-type nsp15 or the corresponding catalytic-deficient nsp15 from human coronaviruses SARS-CoV-1, MERS-CoV, or SARS-CoV-2. At 23 h.p.t, cells were treated with puromycin (5 µg/mL) for 1 h. (A-B) Indirect immunofluorescence analysis was performed to detect nsp15 (magenta), puromycin-labelled *de novo* synthesized peptides (green), and nuclei (DAPI, blue). Fluorescence intensity of nsp15 and puromycin signal along the white line (from a to b) is indicated by histogram plot. Representative images are shown.

relocation of PABPC1 (yellow arrows) was more pronounced in Vero and H1299 cells compared to DF-1 cells (Fig 5B-D). Although there was no clear correlation between the nuclear relocation of PABPC1 and mRNA distribution, an apparent overlap between IBV nsp15 or nsp15-H223A (white arrows) and mRNA (red arrows) was observed in DF-1, H1299, and Vero cells (Figs 5B-D and S3). This suggests the potential for nsp15 to bind to host mRNA.

We proceeded to investigate whether the nuclear relocation of PABPC1 in cells expressing nsp15 from PEDV, TGEV, PDCoV, SARS-CoV-1, MERS-CoV, or SARS-CoV-2 [30] is accompanied by alteration in mRNA distribution. As a control, ARS treatment displayed the migration of PABPC1 and mRNA to both the nucleus and the cytoplasmic stress granules in both PK15 and Vero cells (Fig 6A-B). Similar to the observations with IBV nsp15, overexpression of wild-type nsp15, but not the catalytic-deficient nsp15 of PEDV, TGEV, and PDCoV, resulted in the nuclear accumulation of PABPC1 in PK15 cells (yellow arrows) (Fig 6A). Interestingly, the mRNA signal was noticeably lower (magenta arrows) in nsp15-expressing cells with nuclear relocation of PABPC1 (yellow arrows) compared to cells lacking PABPC1 nuclear relocation, suggesting degradation of cellular mRNA (Fig 6A). Co-localization between catalytic-deficient nsp15 (H226A, H234A) and mRNA was observed (red arrows) (Fig 6A), suggesting the potential interaction of nsp15 and cellular mRNA. In Vero cells, wild-type nsp15 of SARS-CoV-1, but not catalytic-deficient nsp15, led to nuclear localization of PABPC1 (yellow arrows), and this was not accompanied by a decrease in mRNA signal (red arrows) (Fig 6B). In line with the observed effects of MERS-CoV and SARS-CoV-2 nsp15 on protein synthesis, only some

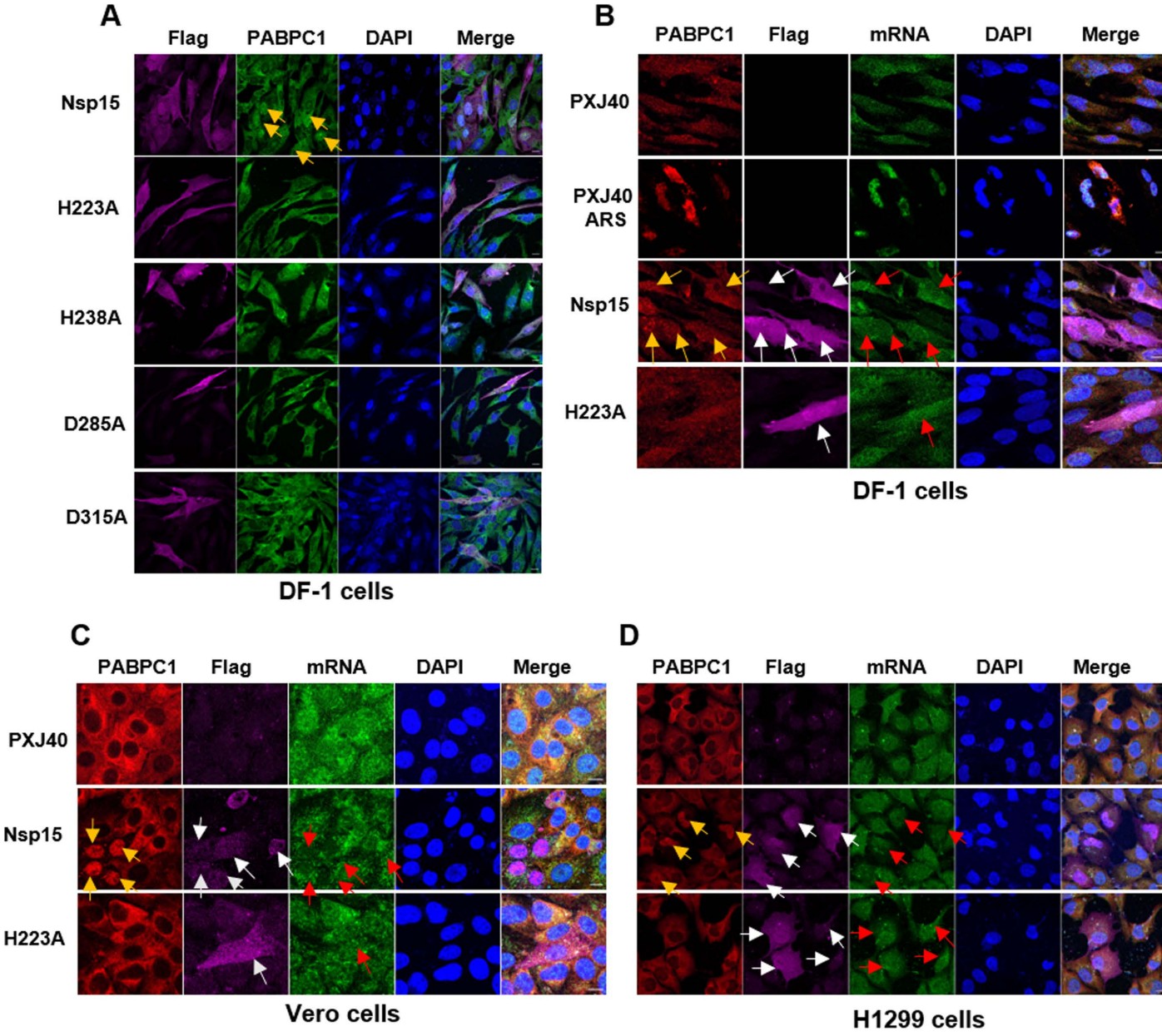

**Fig 5. IBV nsp15 alters PABPC1 localization without affecting cellular mRNA distribution.** (A) DF-1 cells were transfected with plasmids encoding wild-type IBV nsp15 or catalytic/oligomerization-deficient nsp15 mutants (H223A, H238A, D285A, D315A). After 24 h, indirect immunofluorescence staining was performed to visualize nsp15 (magenta) and PABPC1 (green). Nuclei were labelled with DAPI (blue). Cells expressing nsp15 with nuclear accumulation of PABPC1 are indicated by yellow arrows. (B-D) DF-1, Vero, or H1299 cells were transfected with plasmids encoding IBV nsp15 or catalytic-deficient H223A, or the vector PXJ40. After 24 h, mRNA was visualized using oligo dT probes (green) via *in situ* hybridization, followed by indirect immunofluorescence to detect IBV nsp15 (magenta) and PABPC1 (red). Nuclei were labelled with DAPI (blue). Representative images are shown. White arrows indicate the cells expressing IBV nsp15, red arrows indicate the distribution of mRNA, and yellow arrows indicate the cells with PABPC1 nuclear retention. Treatment of DF-1 cells with 1 mM ARS for 30 minutes was used as a positive control for stimulating PABPC1 and mRNA nuclear localization. Representative images from are shown.

of the cells expressing nsp15 of MERS-CoV and SARS-CoV-2 exhibited altered localization of PABPC1 (yellow arrows); while some nsp15-expressing cells, PABPC1 remained in the cytoplasm (blue arrows) (Fig 6B). Once more, co-localization between nsp15 and mRNA signal was observed (red arrows) (Fig 6B), suggesting the binding of nsp15 to mRNA.

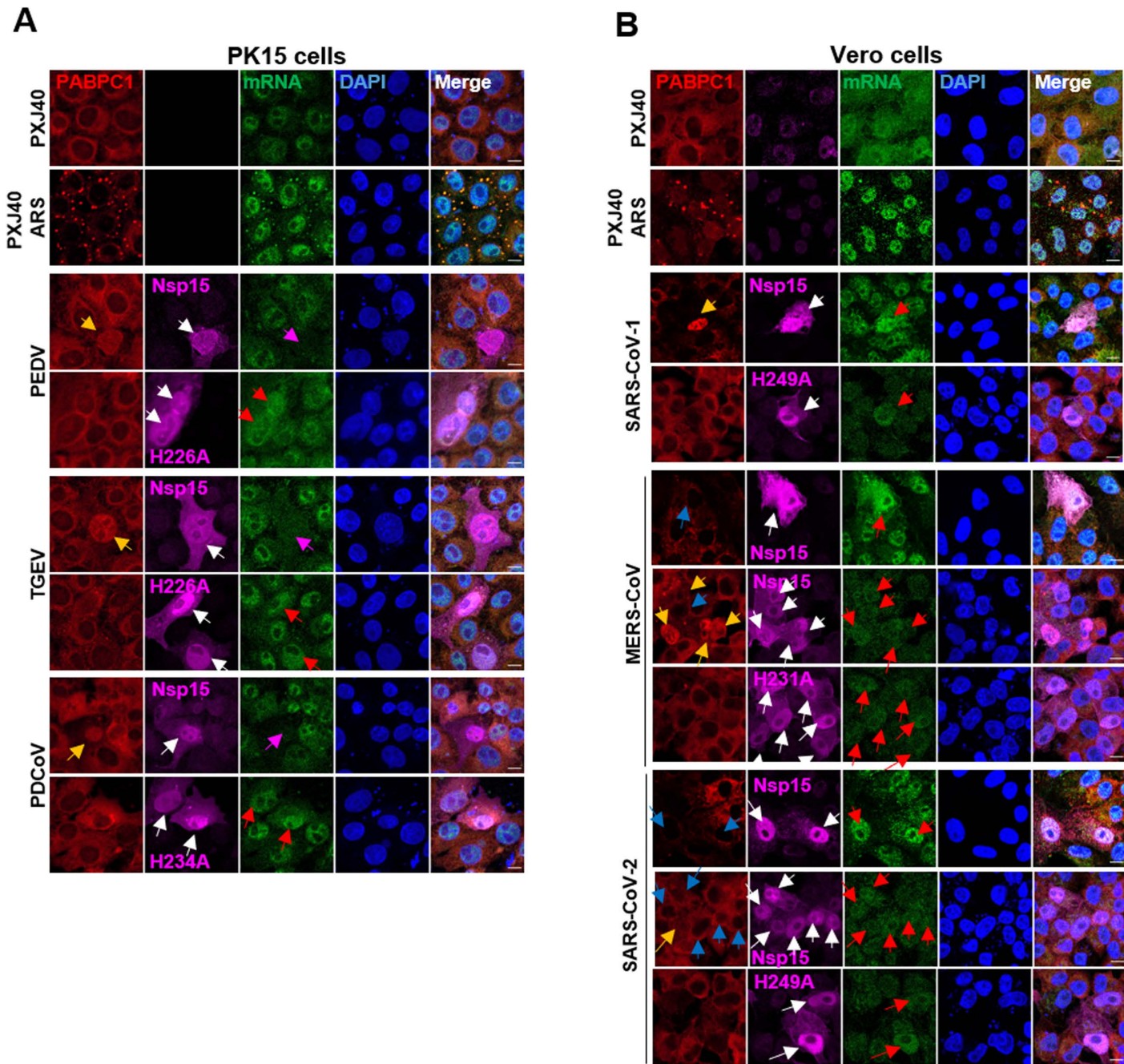

**Fig 6. Nsp15 of PEDV, TGEV, PDCoV, and SARS-CoV-1 alters the localization of PABPC1, while nsp15 of MERS-CoV and SARS-CoV-2 does so in most, but not all, cells.** Wild-type nsp15 or catalytic-deficient nsp15 of the indicated coronaviruses was transfected into the PK15 or Vero cell lines. After 24 h, *in situ* hybridization followed by indirect immunofluorescence was performed to visualize the location of nsp15 (magenta), PABPC1 (red), and mRNA (green). Nuclei were labelled with DAPI (blue). White arrows indicate cells expressing nsp15, yellow arrows indicate cells expressing nsp15 with nuclear localization of PABPC1, blue arrows indicate nsp15 expressing cells without nuclear retention of PABPC1, magenta arrows indicate nsp15-expressing cells with weaker mRNA signal, and red arrows indicate cells with overlapping nsp15 and mRNA distribution. Treatment with 1 mM ARS for 30 min was used as a positive control for visualizing of PABPC1 and mRNA nuclear localization. Representative images are shown.

Taken together, these findings demonstrate that the catalytic activity of nsp15 is required for the nuclear redistribution of PABPC1, indicating that the cleavage of host RNA by nsp15 contributes to PABPC1 nuclear entry. The nuclear localization of PABPC1 in PEDV, TGEV, or PDCoV nsp15-expressing cells is accompanied by a diminished mRNA signal, further suggesting degradation of host mRNA by nsp15. However, the nuclear translocation of PABPC1 mediated by SARS-CoV-1, MERS-CoV, and SARS-CoV-2 nsp15 does not coincide with a decrease in host mRNA levels; colocalization of nsp15 and mRNA is observed, suggesting that these human coronaviruses' encoded nsp15 competitively binds to host mRNA.

## The nuclear localization of PABPC1 induced by nsp15 is accompanied by the inhibition of cellular protein synthesis

Next, we investigated whether the nsp15-mediated relocation of PABPC1 to the nucleus is associated with the shutdown of cellular protein synthesis. As shown in Fig 7A-C, overexpression of wild-type nsp15 from PEDV, TGEV, PDCoV, IBV, and SARS-CoV-1, but not catalytic-deficient nsp15, induced the nuclear localization of PABPC1 (yellow arrows), coinciding with a significant reduction in puromycin signal (red arrows). Consistent with the findings in Fig 6, nuclear localization of PABPC1 was evident in some, but not all, cells expressing nsp15 of MERS-CoV and SARS-CoV-2 (Fig 7C). It was noted that in cells expressing nsp15 of MERS-CoV and SARS-CoV-2, those with nuclear localization of PABPC1

exhibited a weaker puromycin signal (red arrows), whereas cells where PABPC1 localization remained in cytoplasm showed no obvious decrease on the puromycin signal (magenta arrows). As expected, catalytic-deficient mutants of all nsp15 neither altered PABPC1 distribution nor reduced the puromycin signal, underscoring the necessity of catalytic activity for both PABPC1 redistribution and protein synthesis shutdown. These findings establish that nuclear retention of PABPC1 induced by nsp15 correlates with inhibition of *de novo* protein synthesis, however, it remains unclear whether the nuclear translocation of PABPC1 leads to the shutdown of protein synthesis, or whether the shutdown of protein synthesis leads to the nuclear translocation of PABPC1. Notably, SARS-CoV-2 nsp15 appears to be an exception, as it does not evidently inhibit *de novo* protein synthesis in Vero cells.

## IBV nsp15 inhibits protein synthesis in nucleus-free *in vitro* translation system

Next, we sought to determine whether the nsp15-induced protein synthesis shutdown is caused by nsp15 targeting cytosolic or nuclear factors, or both. To achieve this, we employed a nucleus-free *in vitro* translation system, Rabbit Reticulocyte Lysate, to exclude the impact of nuclear factors or PABPC1 nuclear entry. Plasmids encoding nsp15 or the catalytic-deficient mutants H223A and H238A, along with a reporter plasmid encoding IBV N, IBV M, or luciferase, were co-incubated in Rabbit Reticulocyte Lysate for 1 h to allow in vitro translation. Western blot results showed that IBV nsp15 reduced the expression of the reporter plasmids encoding IBV N, IBV M, and luciferase, whereas the catalytic-deficient mutants nsp15 H223A and H238A did not (Fig 8A). These findings suggest that nsp15 can inhibit translation independently of nuclear factors, indicating that the nuclear translocation of PABPC1 is not a necessary step in this process. Instead, it implicates cytosolic factors involved in translation as the likely targets of nsp15. Coupled with evidence showing that nsp15's localization substantially overlaps with that of mRNA, it is plausible that nsp15 directly interacts with cellular mRNA or translation complexes associated with mRNA. Notably, the inhibitory effect of nsp15 on protein synthesis is less pronounced in the *in vitro* translation system compared to its effect observed in cells (Fig 8B). This suggests that nsp15 might also target nuclear factors to induce translation shutdown, a possibility that cannot be entirely ruled out.

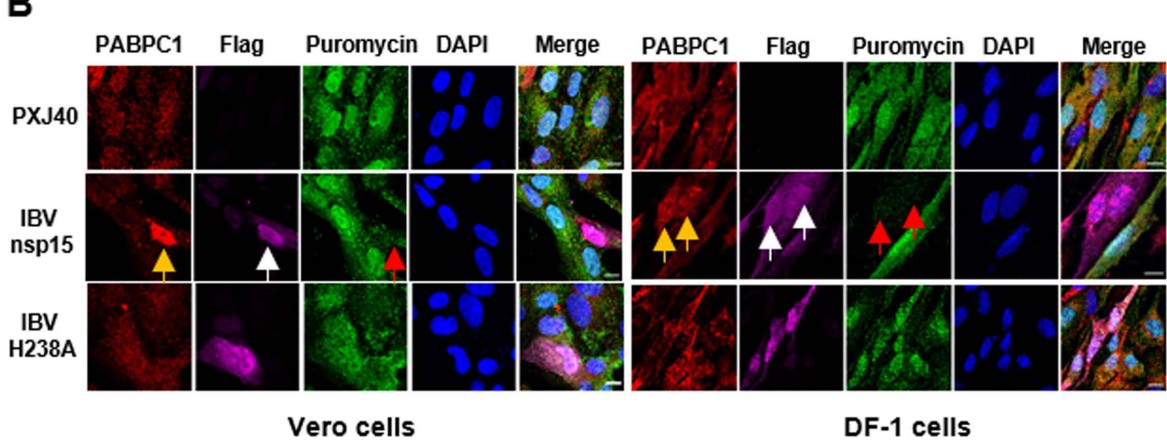

**Fig 7. The nsp15-mediated relocation of PABPC1 is accompanied by inhibition of *de novo* protein synthesis.** (A-C) The PK15 (A), Vero cells and DF-1 cells (B and C) were transfected with PXJ40 or with a plasmid encoding wild type or catalytic-deficient Flag-tagged nsp15 of the indicated coronaviruses. After 23 h, puromycin labelling (5 µg/mL) was performed for 1 h. Indirect immunofluorescence was performed to detect nsp15 (magenta), PABPC1 (red), and puromycin-labelled *de novo* synthesized peptides (green). The nuclei were stained with DAPI (blue). White arrows indicate nsp15-expressing cells, yellow arrows indicate nsp15-expressing cells with PABPC1 nuclear retention, blue arrows indicate nsp15-expressing cells without PABPC1 nuclear retention, red arrows indicate cells with PABPC1 nuclear retention and reduced puromycin labelling signals, and magenta arrows indicate cells expressing nsp15 in which puromycin labelling signals remained stable. Representative images are shown.

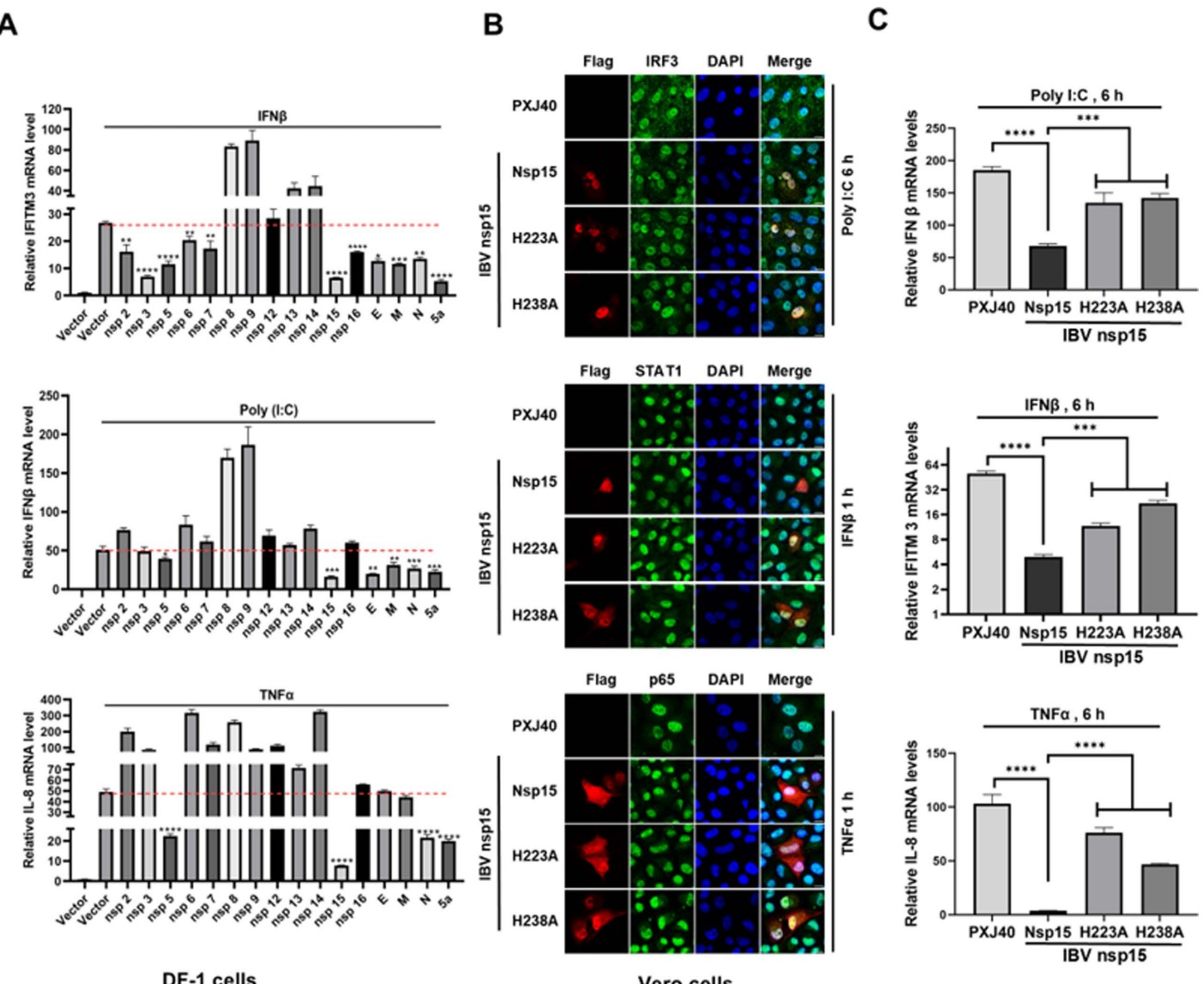

**Fig 8. IBV nsp15 targets cytoplasmic factors to inhibit protein synthesis.** (A) Plasmids encoding wild-type or catalytic-deficient IBV nsp15 and reporter plasmids encoding IBV N or IBV M, or luciferase DNA, were co-incubated with Rabbit Reticulocyte Lysate for 1 h, followed by Western blot analysis or luciferase assay. Density of the bands corresponding to the reporter proteins was normalized to the signal of β-actin and presented relative to the sample transfected with the empty vector PXJ40. ****:$p<0.0001$; *:$p<0.05$. (B) Plasmids encoding wild-type or catalytic-deficient IBV nsp15 and reporter plasmids encoding IBV N or IBV M, or EGFP, were co-transfected into HEK 293T cells for 24 h, followed by Western blot analysis. Band densities of IBV N, IBV M, or EGFP were quantified using ImageJ, normalized to the signal of β-actin, and presented relative to the PXJ40 group.

## Nsp15 decreases the transcriptional levels of antiviral genes induced by various stimuli

During the screening of IBV proteins that antagonize host antiviral gene transcription, we transfected DF-1 cells with plasmids encoding various IBV proteins, followed by stimulation with poly(I:C), IFNβ, and TNFα to induce the expression of IFNβ, ISGs, and IL-8. RT-PCR was then used to detect the transcriptional levels of IFNβ, ISG IFITM3, and IL-8. As shown in Fig 9A, IBV nsp15 exhibited the most significant inhibition of the transcription of anti-viral genes induced

by different stimuli. To check whether the suppression of anti-viral genes transcription by nsp15 is due to the inhibition of nuclear translocation of corresponding transcription factors, we transfected Vero cells with plasmids encoding IBV nsp15 or catalytic-deficient mutant H223A and H238A, and then stimulated the cells with poly(I:C), IFNβ, and TNFα. Immunofluorescence was used to assess the nuclear translocation of IRF3, STAT1, and p65, and real time RT-PCR was used to measure the transcriptional levels of downstream genes IFNβ, IFITM3, and IL-8. As shown in Fig 9B, poly(I:C) stimulation resulted in the nuclear translocation of IRF3, while TNFα stimulation caused the nuclear translocation of p65. The wild type IBV nsp15 and the catalytic-deficient mutants H223A and H238A were unable to inhibit the nuclear translocation of IRF3 and p65, indicating these transcription factors can still initiate the transcription of downstream genes IFNβ and IL-8. It was observed that nsp15 significantly suppressed the transcriptional levels of IFNβ and IL-8, however, the catalytic-deficient mutants H223A and H238A lost this suppression effect (Fig 9C). Since IRF3 and p65 can still translocate into the nucleus, the inhibition of IFN-β and IL-8 expression likely occurs post-transcriptionally.

Examination of the JAK-STAT signaling pathway revealed that STAT1 underwent nuclear translocation in response to IFNβ stimulation. However, in cells expressing IBV nsp15, the STAT1 signal was diminished, and nuclear translocation did not occur (Fig 9B). This was accompanied by a reduction in IFITM3 transcript levels (Fig 9C). In contrast, in cells expressing the catalytic-deficient nsp15 mutants H223A and H238A, STAT1 nuclear translocation continued, and the transcriptional levels of IFITM3 were significantly restored. These results suggest that nsp15 inhibits the JAK-STAT1 signaling pathway, with EndoU activity contributing to this effect. It is also possible that nsp15 directly degrades IFITM3 transcripts. Previous studies have shown that IBV inhibits STAT1 phosphorylation and downstream ISG expression during the late stages of infection [57], our study reveals that nsp15 is involved in this process.

To determine whether the decrease of antiviral gene mRNA levels by nsp15 is conserved across coronaviruses, we transfected PK15 cells with nsp15 from porcine coronaviruses (PEDV, TGEV, PDCoV) and their catalytic-deficient mutants. Similarly, HeLa cells were transfected with nsp15 from human coronaviruses (SARS-CoV-1, MERS-CoV, SARS-CoV-2) and their catalytic-deficient mutants. The cells were then stimulated with poly(I:C), IFNβ, and TNFα. As shown in Fig 10A and 10C, we observed that despite the nuclear translocation of IRF3 and p65, nsp15 from both porcine and human coronaviruses inhibited the transcriptional expression of IFNβ and IL-8, with the EndoU activity playing a crucial role in this process. These results further suggest that coronavirus nsp15 targets IFNβ and IL-8 for degradation or, alternatively, targets other substrates to regulate the transcription levels of these antiviral genes.

Similar to IBV nsp15, these coronavirus-encoded nsp15 also reduced STAT1 signal intensity, inhibited its nuclear translocation, and suppressed the transcription of IFITM3, with EndoU activity being crucial to these effects (Fig 10B). In summary, these results strongly indicate that coronavirus nsp15 likely directly or indirectly targets and degrades antiviral gene transcripts, thereby antagonizing the host's innate immune response. We unexpectedly discovered that nsp15 can reduce STAT1 signal intensity and inhibit its nuclear translocation, with EndoU activity contributing to this effect. This intriguing finding prompts us to further investigate the underlying molecular mechanisms to uncover additional functions of nsp15 in antagonizing the host antiviral response.

## Nsp15 plays a dual role in regulating host protein synthesis during IBV infection

After evaluating nsp15's involvement in regulating host protein synthesis, potentially by targeting cytosolic mRNA and causing the nuclear relocation of PABPC1, we then examined nsp15's impact on host protein synthesis within the context of a viral infection. To achieve

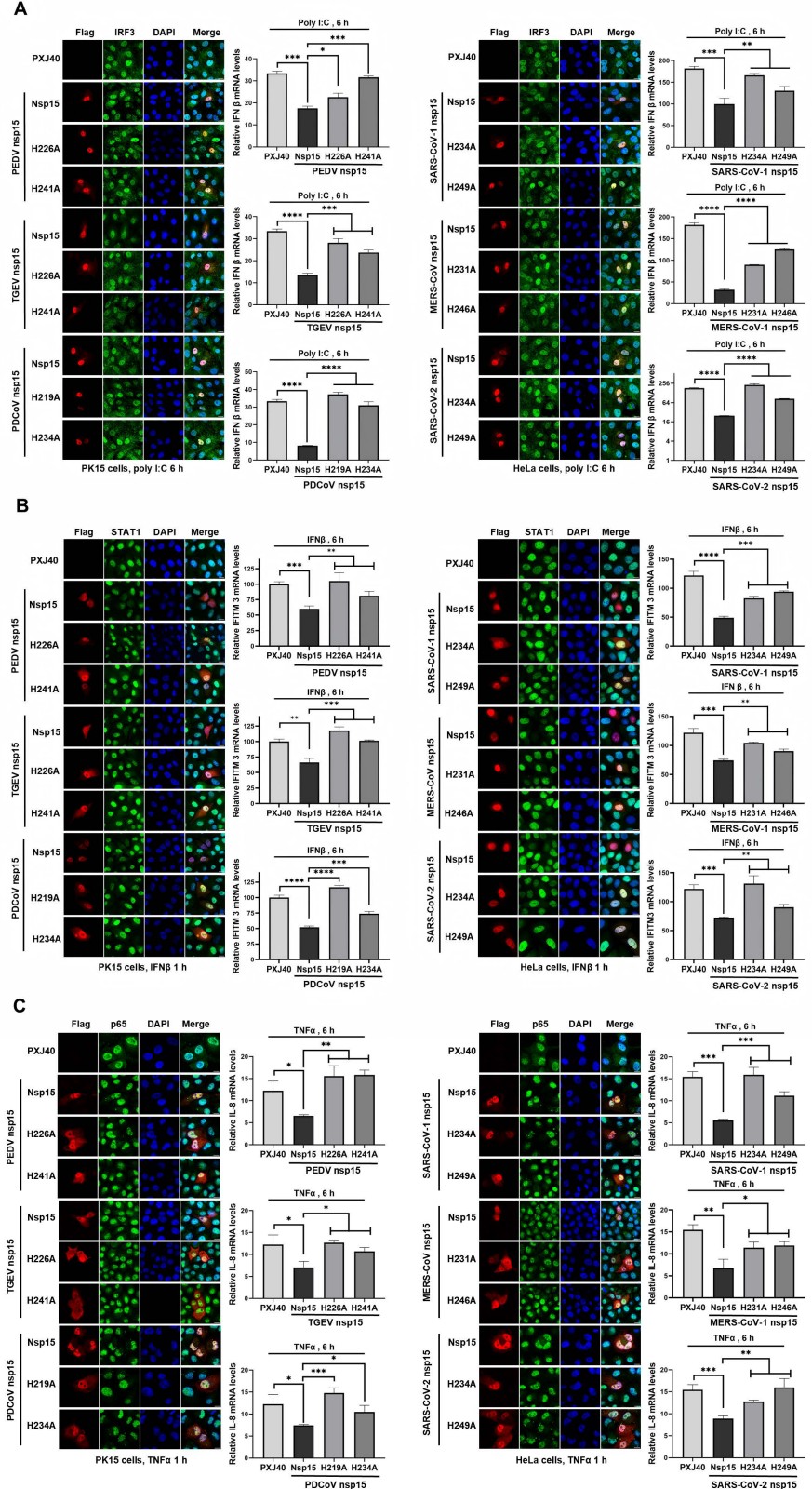

**Fig 9. IBV nsp15 suppresses the transcriptional expression of antiviral genes IFN β, IFITM3, and IL-8.** (A) DF-1 cells were transfected with PXJ40 or plasmids encoding IBV proteins. At 24 h post-transfection, cells were either transfected with poly(**I:**C) or treated with IFNβ or TNFα, followed by real time qRT-PCR analysis to measure the

transcription of IFNβ, IFITM3, and IL-8. (B) Vero cells were transfected with PXJ40 or plasmids encoding IBV nsp15, H223A, or H238A. At 24 h post-transfection, cells were either transfected with poly(I:C) or treated with IFNβ or TNFα. Immunofluorescence was performed to detect the nuclear translocation of IRF3, STAT1, and p65. Representative images are shown. Real time qRT-PCR was also conducted to measure the transcription of IFNβ, IFITM3, and IL-8. Statistical significance levels are denoted as follows: ns, $P > 0.05$; *$P < 0.05$; **$P < 0.01$; ***$P < 0.001$; ****$P < 0.0001$.

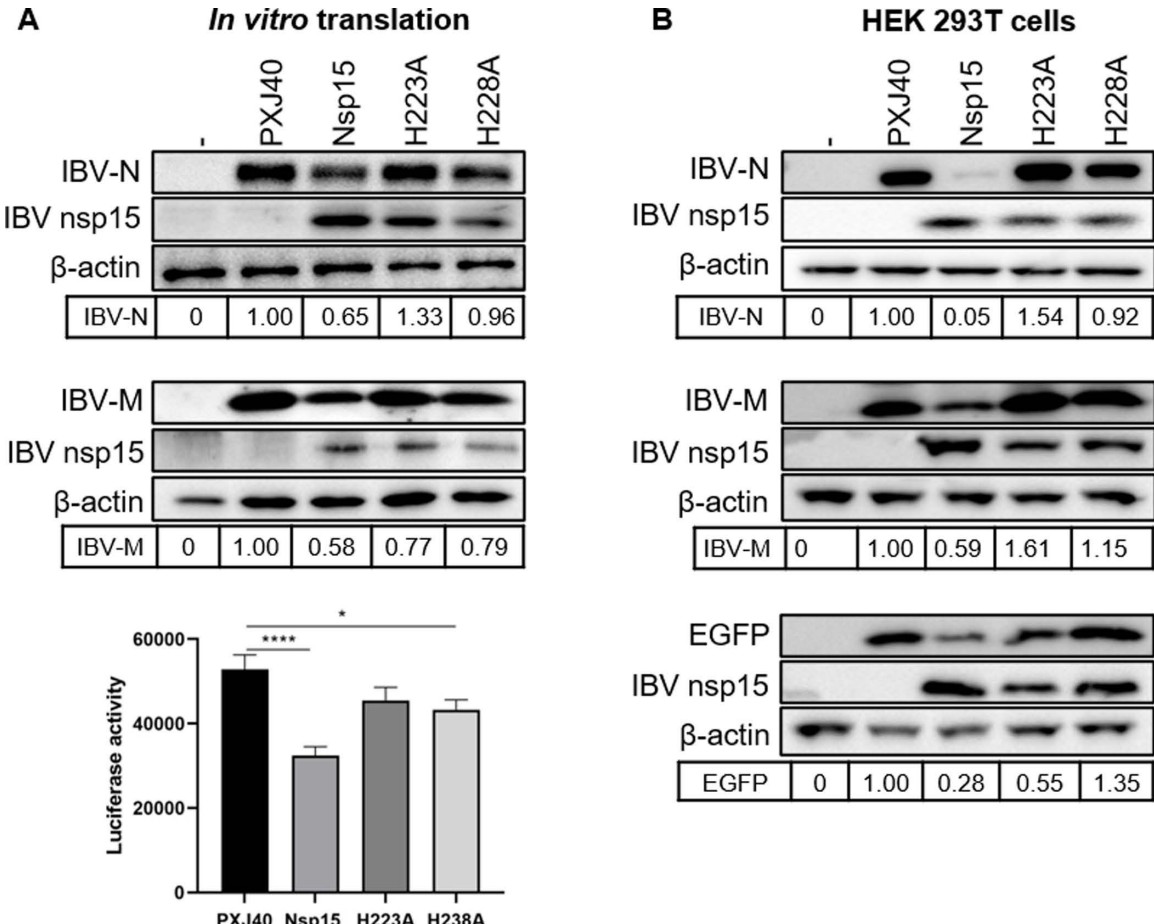

**Fig 10. Coronavirus nsp15 suppresses the transcriptional expression of antiviral genes IFN β, IFITM3, and IL-8.** (A-C) PK15 or HeLa cells were transfected with PXJ40 or plasmids encoding nsp15 or catalytic-deficient mutants from porcine coronavirus (PEDV, TGEV, PDCoV) and human coronavirus (SARS-CoV-1, MERS-CoV, SARS-CoV-2). At 24 h post-transfection, cells were either transfected with poly(I:C) or treated with IFNβ or TNFα. Immunofluorescence staining was performed to detect the nuclear translocation of transcription factors IRF3, STAT1, and p65. Representative images are shown. Real time qRT-PCR was conducted to measure the transcription of IFNβ, IFITM3, and IL-8. Statistical significance levels are denoted as follows: ns, $P > 0.05$; *$P < 0.05$; **$P < 0.01$; ***$P < 0.001$; ****$P < 0.0001$.

this, we utilized a previously reported catalytic-deficient recombinant IBV (rIBV-nsp15-H238A), which contains an alanine substitution in the nsp15 catalytic domain at position H238 [30]. Unexpectedly, Western blot analysis revealed that both wild-type IBV and the rIBV-nsp15-H238A mutant significantly decreased puromycin labeled signals in DF-1 and H1299 cells compared to uninfected controls (Fig 11A-B). This indicates that both forms of the virus can induce a reduction in protein synthesis. Notably, while wild-type IBV caused

translational shutdown, it did not activate the PKR-eIF2α pathway. This was evidenced by low levels of dsRNA until 24 h.p.i. and the absence of increased phosphorylation of dsRNA sensor PKR and the translation initiation factor eIF2α (Fig 11B), consistent with previous studies [30,55,83]. This observation suggests that IBV induces translation shutdown through PKR-eIF2α-independent mechanisms. Interestingly, although the rIBV-nsp15-H238A mutant exhibited reduced replication efficiency, as indicated by lower viral N protein expression, it still reduced *de novo* protein synthesis to a similar extent as wild-type IBV (Fig 11A-B). Consistent with previous findings [30], infection with rIBV-nsp15-H238A led to the accumulation of higher levels of intracellular dsRNA intermediates compared to wild-type IBV. This was associated with increased levels of phosphorylated PKR (p-PKR) and phosphorylated eIF2α (p-eIF2α) (Fig 11B). These results confirm that rIBV-nsp15-H238A triggers a PKR-eIF2α-dependent translational shutdown, which likely impairs both host and viral protein synthesis initiation. This host-mediated translation initiation checkpoint shutoff triggered by rIBV-nsp15-H238A apparently does not benefit viral replication.

To further confirm nsp15's role in cleaving viral RNA, we conducted a Northern Blot analysis to compare the levels of total viral RNA in cells infected with either wild-type IBV or the rIBV-nsp15-H238A mutant. The results revealed that cells infected with rIBV-nsp15-H238A accumulated significantly higher levels of viral RNA compared to those infected with wild-type IBV (Fig 11C). This finding demonstrates that nsp15 is crucial in facilitating viral redundant RNA degradation, allowing the virus to evade the PKR-eIF2α-dependent translation shutdown and the subsequent stress response, which would otherwise hinder viral replication.

Besides bypassing the PKR-eIF2α-mediated translation shutdown, which is detrimental to viral protein synthesis, nsp15 may inhibit protein synthesis through additional mechanisms. To rule out the influence of the PKR-eIF2α pathway on translation inhibition, we silenced PKR expression in H1299 cells using siRNA transfection. As shown by the western blot in Fig 11D, wild-type IBV infection led to similar levels of translation inhibition in both PKR knockdown cells and PKR-positive cells. In contrast, rIBV-nsp15-H238A infection caused less inhibition of protein synthesis in PKR knockdown cells compared to PKR-positive cells. Moreover, when comparing translation efficiency between wild-type IBV-infected and rIBV-nsp15-H238A-infected PKR knockdown cells, we found that rIBV-nsp15-H238A induced less translation inhibition than wild-type IBV (Fig 11D, right panel). These findings demonstrate that nsp15 helps the virus to inhibit host protein synthesis through a PKR-eIF2α-independent mechanism, while also allowing the virus to evade the PKR-eIF2α-mediated translation initiation shutdown, which would otherwise impair viral protein production.

**Infection with rIBV-nsp15-H238A, but not with wild-type IBV, induces nuclear retention of both PABPC1 and mRNA**

As we observed a correlation between PABPC1 nuclear retention and inhibition of *de novo* protein synthesis only in cells overexpressing wild-type nsp15, but not the catalytic-deficient nsp15 (Fig 7), we next examined the localization of PABPC1 in cells infected with wild-type IBV and rIBV-nsp15-H238A. Contrary to the nuclear retention of PABPC1 observed in nsp15 overexpressing cells, PABPC1 did not undergo nuclear translocation in most wild-type IBV-infected cells (Fig 12A, left panel). FISH analysis revealed that upon infection with wild-type IBV, mRNA is evenly distributed throughout the cytoplasm in both Vero and DF-1 cells, without nuclear retention (Fig 12B, white circle; Fig 12C, white arrows). These results suggest that, despite the occurrence of cellular translation shutoff, PABPC1 binds to viral mRNA to facilitate its translation, thereby remaining in the cytoplasm. However, in some but not all rIBV-nsp15-H238A-infected cells, nuclear localization of PABPC1 was observed from 12 h.p.i. onwards (Fig 12A, right panel, white circles). This is inconsistent with the phenomenon observed in cells overexpressing nsp15-H238A, where PABPC1 does not enter the nucleus

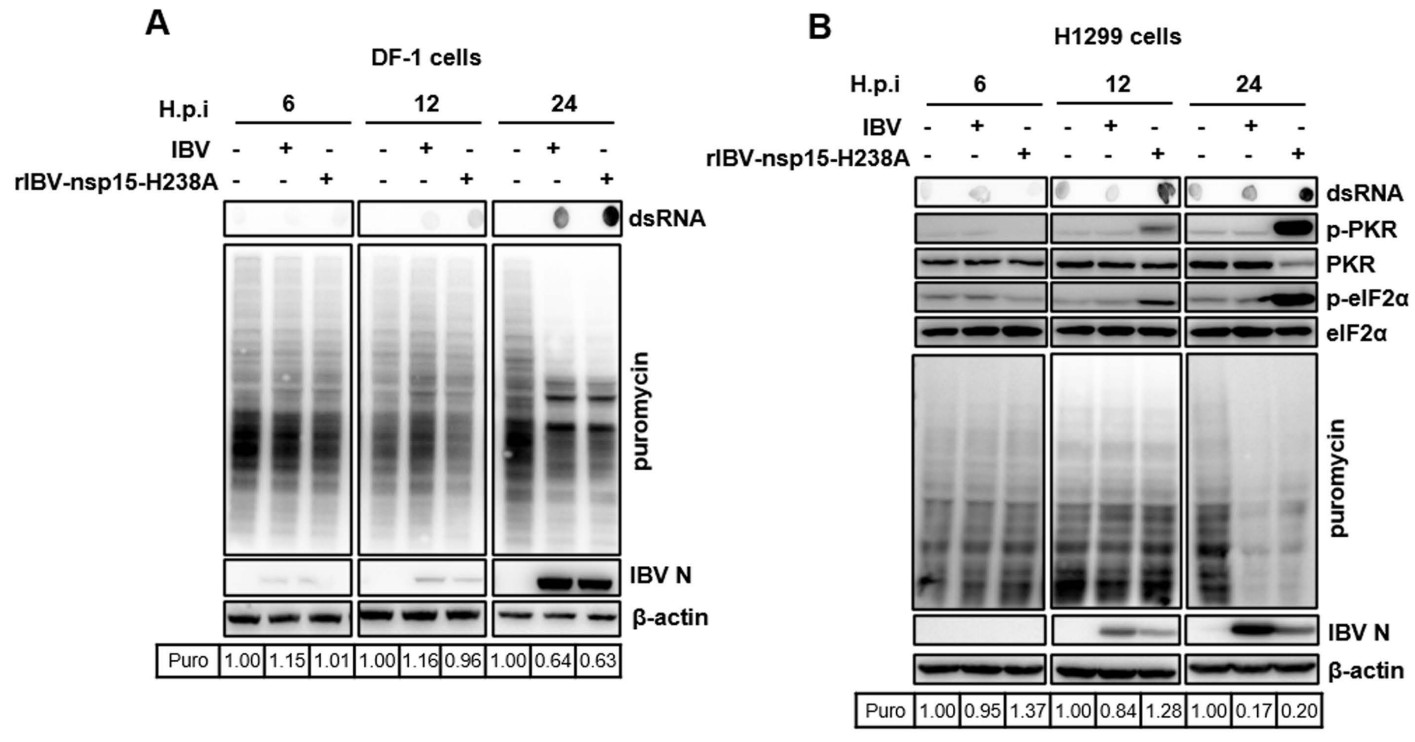

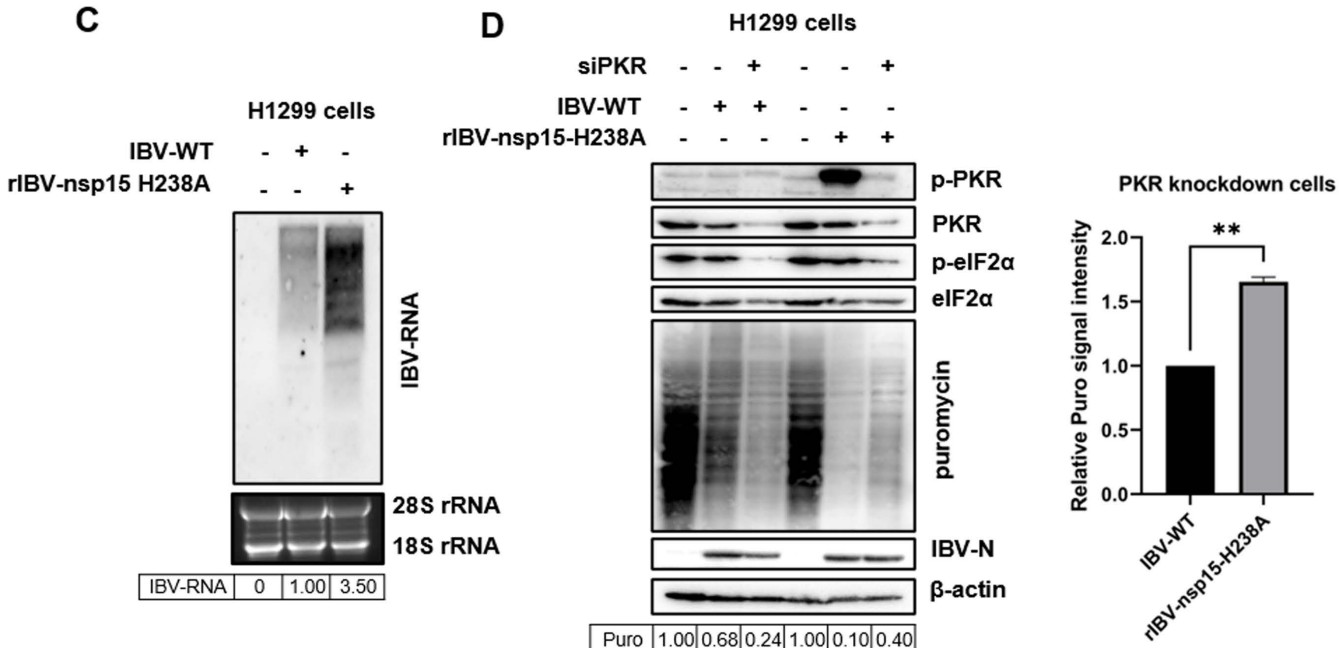

**Fig 11. Nsp15 plays a dual role in regulating host protein synthesis efficiency during IBV infection.** (A) DF-1 cells or (B) H1299 cells were infected with wild-type IBV or rIBV-nsp15-H1238A at an MOI of 1. At 6, 12, 24 h.p.i., cells were treated with puromycin (5 μg/mL) for 1 h, followed by western blot analysis to detect puromycin-labelled *de novo* peptides, IBV-N, p-PKR, PKR, p-eIF2α, eIF2α, and β-actin. Dot blot analysis was performed to detects dsRNA. (C) H1299 cells were infected with wild-type IBV or rIBV-nsp15-H1238A for 18 h, followed by Northern blot analysis to detects viral RNA. (D) H1299 cells were transfected with siRNA targeting PKR or control siRNA for 36 h, followed by infection with IBV or rIBV-nsp15-H1238A for 24 h and incubation with puromycin for 1 h. Subsequently, cells were subjected to western blot analysis to detect p-PKR, PKR, p-eIF2α, eIF2α, puromycin-labelled *de novo* peptides, IBV-N, and β-actin.

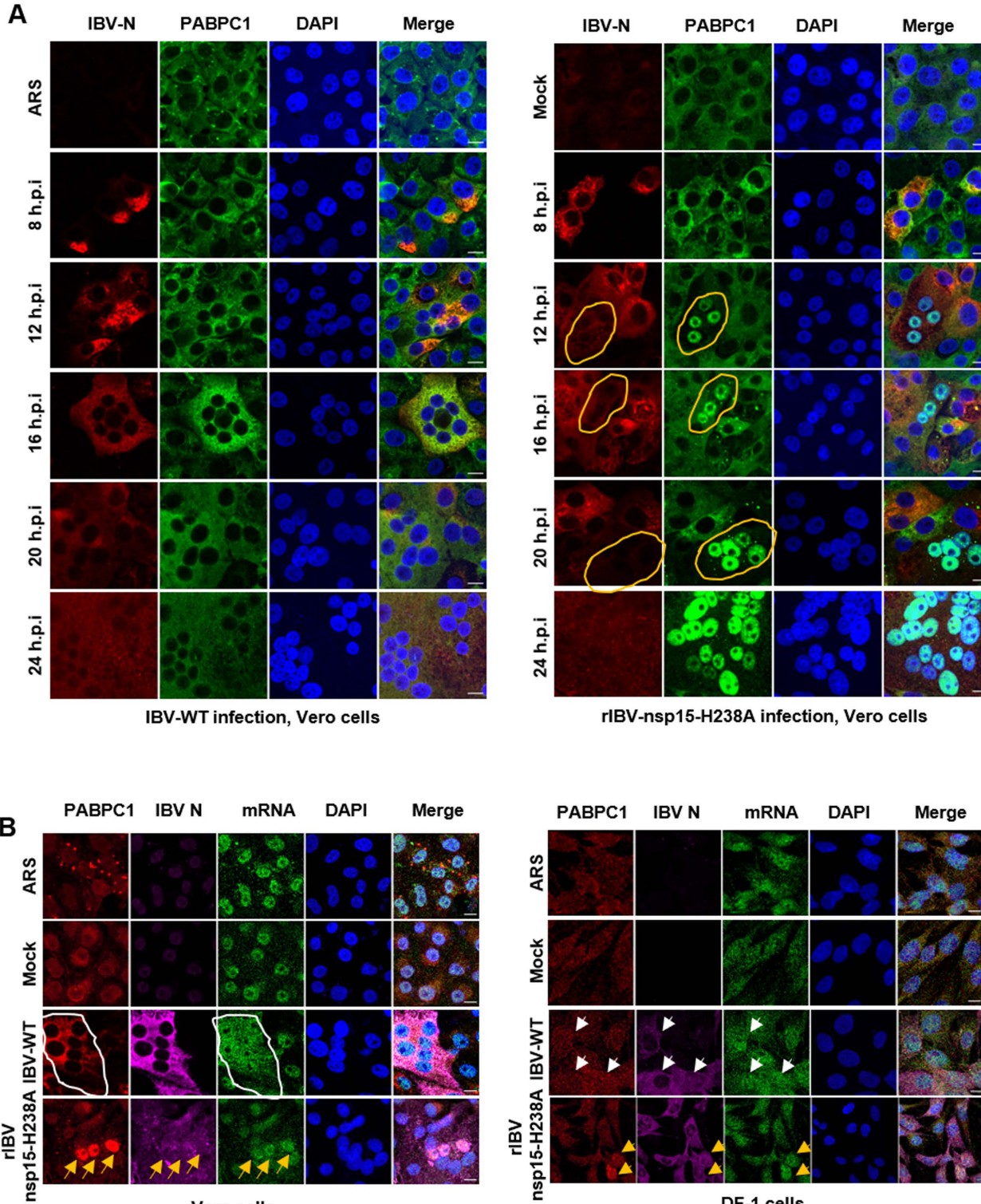

**Fig 12. Infection with rIBV-nsp15-H238A, but not with wild-type IBV, induces nuclear retention of both PABPC1 and mRNA, and this is associated with reduced viral protein synthesis.** (A) Vero cells were infected with wild-type IBV or treated with ARS (left panel), or infected with rIBV-nsp15-H1238A (right panel). At 8, 12, 16, 20 and 24 h.p.i, indirect immunofluorescence was performed to detect IBV-N protein (red) and PABPC1 (green). Nuclei were stained with DAPI (blue). White circles indicate the rIBV-nsp15-H238A-infected cells with nuclear localization of PABPC1 and weaker IBV-N signals. (B) Vero and DF-1 cells were infected with wild-type IBV or rIBV-nsp15-H1238A. At 16 h.p.i., FISH and

indirect immunofluorescence were performed to detected PABPC1(red), IBV N (magenta), and mRNA (green). Nuclei were stained with DAPI (blue). The white circle and white arrows indicate wild-type IBV-infected cells in which no nuclear accumulation of PABPC1 and re-distribution of mRNA were observed, while the yellow arrows indicate rIBV-nsp15-H238A-infected cells that exhibit nuclear localization of both PABPC1 and mRNA. Representative images are shown.

(Fig 5A). This discrepancy may be attributed to the PKR-eIF2α signaling pathway-dependent translation shutoff induced by rIBV-nsp15-H238A infection. This pathway prevents the initiation of translation for both host and viral mRNAs, leading to the release of PABPC1 from the mRNA and exposing its nuclear localization signals, which directs it to the nucleus. The PKR-eIF2α-mediated translation shutoff is detrimental to viral replication, as indicated by the weaker IBV N protein signal observed in rIBV-nsp15-H238A-infected cells with nuclear retention of PABPC1 compared to those without PABPC1 nuclear retention (Fig 12A, right panel, white circles). FISH combined with indirect immunofluorescence showed that in wild-type IBV-infected cells, there was no nuclear retention of PABPC1 or mRNA (Fig 12B, white circles in Vero cells and white arrows in DF-1 cells). Conversely, in cells infected with rIBV-nsp15-H238A, both PABPC1 and mRNA were retained in the nucleus (Fig 12B, yellow arrows). This finding reveals that the PKR-eIF2α pathway-mediated translation shutoff causes PABPC1 to accumulate in the nucleus, leading to the obstruction of mRNA nuclear export and further sequestration of host mRNA in the nucleus.

## IBV nsp15 is associated with the replication-transcription complex (RTC) and binds with viral RNA and host RNA during infection

Subcellular localization is crucial for a protein's interaction with its partners and provides key insights into its function. To investigate the localization of IBV nsp15, we examined its distribution both following plasmid transfection and during IBV infection in DF-1 cells. Indirect immunofluorescence revealed that exogenous Flag-tagged nsp15 primarily localized to the cytoplasm, with some presence in the nucleus (Fig 13A, top panel). In contrast, infection with wild-type IBV or rIBV-nsp15-H238A resulted in the formation of perinuclear aggregates of nsp15 or nsp15-H238A (Fig 13A, middle and bottom panels). The observed co-localization of nsp15 with the RTC core protein RdRp nsp12 (Fig 13B) supports the association of nsp15 with the RTC during infection. These findings highlight that nsp15 exhibits distinct localization patterns depending on the context: in plasmid-transfected cells, nsp15 is more diffuse throughout the cytoplasm and nucleus, likely due to the absence of viral RNA and other viral proteins. In contrast, during virus infection, nsp15 interacts with other viral proteins and RNA, localizing to the RTC to manage viral dsRNA levels and support genome replication and transcription by balancing positive and negative RNA strands.

To investigate the RNA-binding ability of IBV nsp15, H1299 cells were infected with IBV, followed by nsp15-RNA immunoprecipitation sequencing (RIP-Seq) analysis. Western blot showed the successfully immunoprecipitation of nsp15 (Fig 13C). Sequencing analysis showed that nsp15 specifically enriched 237 cellular RNAs and 6 viral RNAs (S1 Data), after excluding RNAs enriched in the control group (IgG). Six viral RNAs include full length genomic RNA/mRNA1 and the subgenomic mRNAs (mRNA2, mRNA3, mRNA4, mRNA5, mRNA6), which encoding the polyprotein 1a and 1ab, structural proteins and accessary proteins S, 3a, 3b, E, M, and N. This interaction implies that nsp15 may play a direct role in modulating the stability, processing, or translation of viral transcripts during replication cycle. Gene Ontology (GO) analysis of the 237 cellular RNAs associated with nsp15 revealed significant enrichment in biological processes (BP) included cytoplasmic translation, translation, peptide biosynthetic process and metabolic process, and amide biosynthetic process, underscoring nsp15's role in

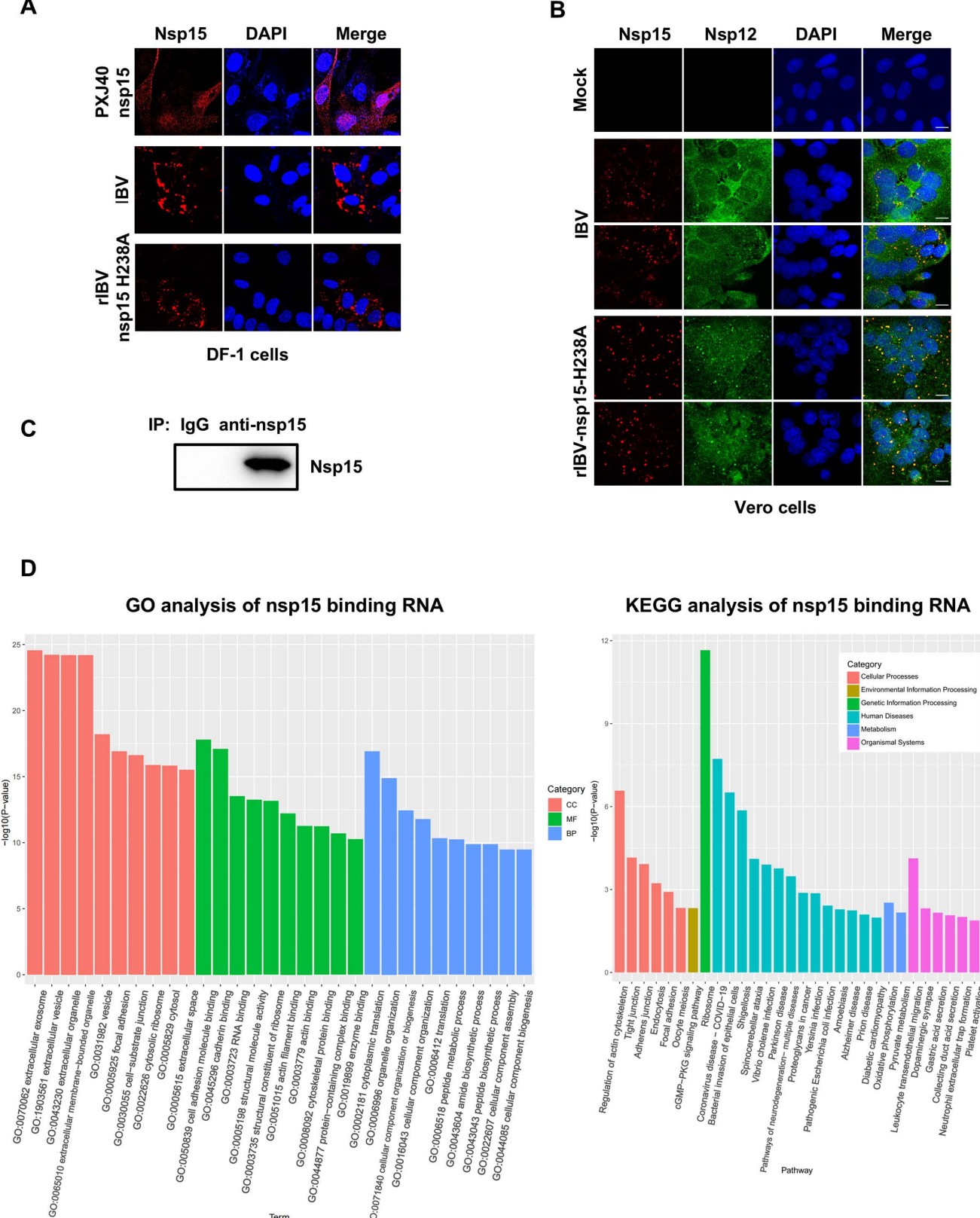

**Fig 13. IBV nsp15 is associated with the RTC major component nsp12.** (A) DF-1 cells were transfected with a plasmid encoding Flag-tagged nsp15 or infected with either IBV or rIBV-nsp15-H238A. At 24 h.p.t. or 18 h.p.i., indirect immunofluorescence was performed with anti-Flag antibody or with

mouse anti-IBV-nsp15 monoclonal antibody (red). Nuclei were stained with DAPI (blue). (B) Vero cells were infected with IBV or rIBV-nsp15-H238A. At 18 h.p.i, indirect immunofluorescence was performed with mouse anti-IBV-nsp15 monoclonal antibody and rabbit anti-IBV-nsp12 polyclonal antibody. The nuclei were stained with DAPI (blue). (C) H1299 cells were infected with wild-type IBV for 24 h. Whole-cell lysates were subjected to immunoprecipitation using an anti-nsp15 monoclonal antibody or a Mouse IgG antibody as a control, followed by RNA extraction. Western blot was performed to confirm the expression and successful immunoprecipitation using anti-nsp15 monoclonal antibody. RNA yields were quantified using a NanoDrop spectrophotometer, and the eluted RNA was analyzed via high-throughput sequencing. (D) The left panel presents the enrichment of GO annotations for proteins encoded by nsp15-associated RNAs, while the right panel highlights the enrichment of KEGG pathways for these proteins.

translation regulation. Additionally, terms related to the biogenesis, assembly, and organization of cellular components were prominently enriched, suggesting broader functional implications (Fig 13D, left panel). In the cellular component (CC) category, the nsp15-associated RNAs linked to the extracellular exosome, extracellular vesicle, extracellular organelle, focal adhesion, cell-substrate junction, and cytosolic ribosome. This indicates that nsp15-associated RNAs are distributed across diverse subcellular compartments. In the molecular function (MF) category, enriched terms included RNA binding and structural constituent of the ribosome, reflecting nsp15's interaction with RNAs involved in RNA metabolism and translation. Kyoto Encyclopedia of Genes and Genomes (KEGG) pathway enrichment analysis revealed significant involvement of nsp15-associated RNAs in various cellular and biological pathways. The most enriched pathway was ribosome, highlighting nsp15's impact on protein synthesis. Other enriched pathways included coronavirus disease (COVID-19), emphasizing nsp15's specific role in viral replication and pathogenesis (Fig 13D, right panel). Together, these findings demonstrate that nsp15 interacts with both viral and host RNAs, playing a crucial role in regulating the viral life cycle and host gene expression.

## The nsp15 interactome is mainly involved in RNA processing and protein translation

To further investigate whether nsp15 targets host translation-related proteins to inhibit protein synthesis, we infected H1299 cells with IBV. Immunoprecipitation was performed using nsp15-specific monoclonal antibody to pull down nsp15 and its interacting proteins, followed by mass spectrometry analysis, which identified a total of 809 interacting proteins (S2 Data). These nsp15 interactors include numerous proteins associated with transcription and translation, such as rRNA processing proteins, ribosome biogenesis factors, ribosomal components, eukaryotic initiation factors (eIF2S1-3, eIF3A-M, eIF4A1-3, eIF4G1-3), and RNA-binding proteins (e.g., PABPC1, PABPN1, PABPC4, hnRNP, RBM, and others). GO enrichment analysis of the nsp15-interacting proteins revealed significant involvement in biological process included mRNA splicing, RNA splicing, mRNA processing, rRNA processing, translation, cytoplasmic translation, mitochondrial translation, regulation of translation initiation, positive regulation of translation, and ribosomal small subunit biogenesis (Fig 14), highlighting nsp15's potential role in regulating RNA metabolism and protein synthesis. Furthermore, terms related to protein folding, intracellular protein transport, ER to Golgi transport, chaperone-mediated ubiquitin-dependent processes, protein stabilization, and proton motive-driven mitochondrial processes, suggests that nsp15 plays a broad role in modifying the host cell's protein homeostasis, protein transport, and cellular energy balance during infection. In the cellular component category, the interacting proteins were predominantly associated with the cytosol, nucleus, mitochondria, and extracellular exosomes, indicating nsp15's distribution across various subcellular compartments. Notably, the enrichment of nucleoplasm and ribosomal components suggests its involvement in transcriptional and translational regulation. For molecular function, the enriched terms were dominated by RNA binding, mRNA

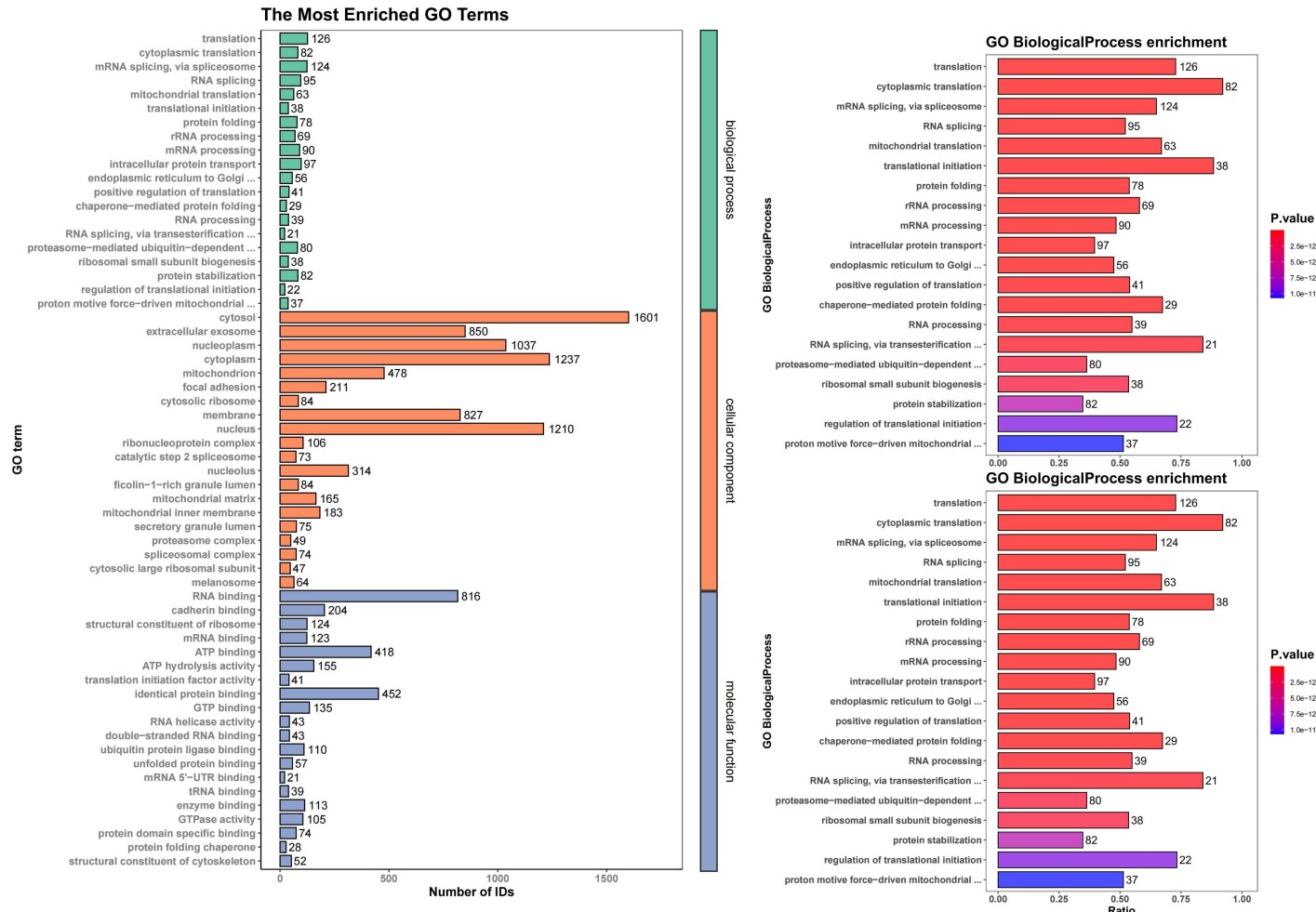

**Fig 14. H1299 cells were infected with wild-type IBV for 24 hours.** Whole-cell lysates were subjected to immunoprecipitation using an anti-nsp15 antibody, with a Mouse IgG antibody serving as the control. Immunoprecipitated proteins were subsequently analyzed by mass spectrometry. The results display the enrichment of Gene Ontology (GO) annotations and KEGG pathways associated with nsp15-interacting proteins.

binding, RNA helicase activity, dsRNA binding, mRNA 5'-UTR binding, tRNA binding, structural constituent of the ribosome, translation initiation factor activity, demonstrating that nsp15 plays a significant role in RNA metabolism and protein translation during infection. The enriched terms such as ubiquitin protein binding, unfolded protein binding, protein folding chaperone, suggests that nsp15 may play a critical role in regulating protein quality control and maintaining proteostasis. Additionally, terms such as ATP binding, ATP hydrolysis activity, GTP binding, and GTPase binding were also enriched, implying that nsp15 could play a role in regulating cellular energy states or influencing key cellular pathways that require ATP or GTP for their function. These findings suggest that nsp15 may exert its effects on host protein synthesis by interacting with proteins involved in RNA metabolism, translation, and protein quality control, providing valuable insights into its functional mechanisms.

The KEGG pathway enrichment analysis revealed that nsp15-interacting proteins are significantly associated with multiple critical cellular and viral pathways (Fig 14). Among these, the most enriched pathways include ribosome, spliceosome, mRNA surveillance, and aminoacyl-tRNA biosynthesis, highlighting nsp15's involvement in RNA splicing, processing,

and quality control, and translation. Additionally, pathways related to protein processing in the ER and proteasome suggest its role in protein folding and proteostasis. Interesting, the coronavirus disease-COVID-19 pathway was enriched, emphasizing the relevance of these interactions in coronavirus pathogenesis. Other notable pathways include the enrichment of the TCA cycle and oxidative phosphorylation pathways, suggest that nsp15 may influence host cellular energy metabolism, specifically in the context of mitochondrial function and ATP production. In summary, these findings reveal that nsp15 exerts a broad influence on host cellular processes, including RNA splicing, translation, protein folding and transport, protein quality control, and energy metabolism.

## Discussion

Several viruses encode ribonuclease and employ fine-tuned tactics to control viral as well as host mRNA expression, balancing viral and host protein expression [74,84–86]. The conserved EndoU nsp15 is the unique genetic marker of *Nidovirales*, as it is not present in other RNA virus families. In our previous study, nsp15 from various genera of coronaviruses was found to suppress stress granule formation, either by impeding the accumulation of viral dsRNA or by targeting unknown host factors [30]. Thus, the role of nsp15 has been revealed to be more intricate than initially assumed, owing to its potential to act on both viral and cellular substrates. While the influence of nsp15 on the regulation of viral RNA is well documented

[27,28,30,87], the cellular targets of nsp15 remain undisclosed. In this study, through the transfection of nsp15-encoding plasmids into eukaryotic cells, we demonstrate that nsp15 from various genera of coronaviruses hampers global protein synthesis, concomitant with the re-localization of PABPC1 to the nucleus. Additionally, the transcripts of antiviral genes are reduced in nsp15 expressing cells. These effects were largely nullified upon expression of catalytic-deficient nsp15, highlighting the crucial role of EndoU activity in the shutdown of protein synthesis and evasion of the innate immune response. When catalytic-deficient nsp15-H238A was introduced into the avian coronavirus IBV genome, we found that during infection, compared to catalytic-deficient rIBV-nsp15-H238A, wild-type IBV with functional nsp15 could reduce the accumulation of viral RNA, avoiding the activation of the PKR-eIF2α signaling pathway and escaping p-eIF2α-mediated global translation shutoff that is detrimental to both host and viral protein synthesis. Meanwhile, IBV nsp15 binds to 257 cellular RNAs and 809 cellular proteins, contributing to the regulation of RNA splicing, processing, translation, thereby inhibiting host protein synthesis in a PKR-eIF2α-independent mechanism. This represents the inaugural report implicating coronavirus-encoded EndoU in the modulation of host gene expression.

Protein synthesis initiation plays a pivotal role in determining the efficiency of both host and viral protein synthesis [88]. PABPC1, a nucleocytoplasmic shuttling protein, predominantly resides in the cytoplasm where it facilitates translation initiation by concurrently binding to the mRNA poly(A) tail and the eukaryotic initiation factor 4F complex (eIF4F). The eIF4F complex attaches to the 5' cap of mRNA and is integral to the translation complex (TC). Following the delivery of mRNA-TC to ribosomes, PABPC1 dissociates and can associate with the importin α/β complex, facilitating the nuclear import of PABPC1 [81,89–91]. In response to various pathogenic and non-pathogenic stressors, PABPC1 undergoes nuclear relocation or aggregates into stress granules [81,92–94]. Elevated levels of PABPC1 in nucleus leads to the addition of longer poly(A) tails to mRNAs and the hyperadenylated mRNAs will be retained within the nucleus rather than being exported to the cytoplasm [95], thereby preventing their translation. In this study, we observed nuclear relocation of PABPC1 associated with protein synthesis shutdown in all cells overexpressing nsp15. Given PABPC1's capability to

shuttle between the nucleus and cytoplasm [80], and previous reports indicating that blocking mRNA export from the nucleus to the cytoplasm typically results in nuclear retention of PABPC1 [81], we investigated whether the nuclear relocation of PABPC1 associated with nsp15 is due to mRNA nuclear retention or enhancement of nuclear import. *In situ* hybridization detecting poly(A) mRNA transcripts did not display nuclear retention of mRNA in nsp15-expressing cells. This indicates that EndoU nsp15 does not block mRNA nuclear export to retain PABPC1 in the nucleus. Therefore, nuclear relocation of PABPC1 is likely the result of enhanced nuclear import. Nuclear import of PABPC1 depends on its interaction with the importin-α/β complex [81]. The motifs of PABPC1 that bind to the importin-α/β complex are the same ones that recognize and bind mRNA. Dissociation from cytoplasmic mRNA would expose PABPC1's nuclear import signal, facilitating its shuttling to the nucleus [81,96]. The binding of nsp15 to cellular mRNA potentially competes with PABPC1, as indicated by the general co-localization of nsp15 with cellular mRNA, thereby releasing PABPC1 from cytoplasmic mRNA and promoting its nucleus entry. In PK15 cells expressing nsp15 of porcine coronaviruses (TGEV, PEDV, PDCoV), a reduction in cellular mRNA signal was observed, suggesting that nsp15 from some coronaviruses may not only bind but also degrade host mRNA. Consequently, the relocation of PABPC1 to the nucleus, likely prompted by the competition for cellular mRNA binding and/or mRNA degradation by nsp15, may therefore be a consequence rather than the cause of translation shutoff. This notion is further supported by nucleus-free *in vitro* translation studies, demonstrating that the inhibition of protein synthesis by nsp15 is not reliant on PABPC1 nucleus entry.

To evaluate whether nsp15 mediates host protein synthesis shutdown also during virus infection, we infected cells with wild-type IBV and the catalytic-deficient rIBV-nsp15-H238A. Interestingly, we observed that both viruses inhibited host protein synthesis but via different mechanisms. Wild-type IBV inhibited host protein expression in an eIF2α checkpoint-independent manner, likely through a yet unknown virus-mediated mechanism. On the other hand, rIBV-nsp15-H238A triggered host protein shutdown in a manner dependent on the activation of the dsRNA-PKR-eIF2α pathway, attributed to the accumulation of higher levels of dsRNA during infection with this EndoU catalytic-deficient virus. Interestingly, we did not observe nuclear retention of PABPC1 in cells infected with wild-type IBV, whereas PABPC1 nucleus accumulation was clearly evident in cells infected with rIBV-nsp15-H238A. This apparent contradiction regarding PABPC1 re-localization between transfection and infection conditions could be explained by the presence of viral RNA during infection. Infection with IBV leads to eIF2α checkpoint-independent protein synthesis shutdown, in this manner, viral mRNAs' poly(A) tails may still interact with cytoplasmic PABPC1 for their translation [97], thereby retaining PABPC1 in the cytoplasm. In contrast, infection with rIBV-nsp15-H238A results in the accumulation of viral dsRNA due to the loss of nsp15 EndoU activity. This accumulation triggers the activation of the PKR-eIF2α pathway, leading to the shutdown of global mRNA translation. Consequently, PABPC1 is released from host/viral mRNA and translocated into the nucleus. This hypothesis is supported by previous studies demonstrating that the low abundance of cytoplasmic mRNA releases PABPC1 into the nucleus [63,96,98]. Intriguingly, in cells infected with rIBV-nsp15-H238A, the nuclear localization of PABPC1 was accompanied by a lower IBV N protein signal compared to infected cells without PABPC1 nuclear relocation. This could be attributed to the stalled translation initiation of viral mRNA, which leads to the release of more PABPC1, allowing it to enter the nucleus. Northern blot analysis confirms that rIBV-nsp15-H238A accumulates more viral RNA than wild-type IBV (Fig 11C), further demonstrating that nsp15 targets viral RNA for cleavage. Although more viral RNA accumulates, Western blot analysis shows that this mutant virus produces less viral protein (Fig 11A-B), implying that the accumulated viral RNA is not efficiently translated.

Therefore, rIBV-nsp15-H238A induces translation shutdown through the accumulation of dsRNA and activation of the PKR-eIF2α pathway, impede the initiation step of translation (Met-tRNA recruitment), effectively halting both host and viral protein synthesis. This leads to reduced viral replication, as evidenced by the decreased expression of the viral N protein. Therefore, the PKR-eIF2α-dependent translational shutdown, together with the induction of type I interferon [30], is responsible for the lower replication of rIBV-nsp15-H238A.

The eIF2α checkpoint-induced host shutoff inhibits both host and viral protein synthesis and serves as a host defense mechanism against viruses [99,100]. In contrast, eIF2α-independent translation shutoff might specifically regulate host protein expression [71]. To confirm the role of nsp15 in virus induced PKR-eIF2α-independent translation shutoff, we knocked down PKR in H1299 cells, to exclude the PKR-eIF2α-mediated translation shutdown. In PKR positive cells, both wild-type IBV and rIBV-nsp15 infection result in protein synthesis shutdown; however, in PKR knockdown cells, wild-type IBV still retains the capacity to reduce host protein synthesis, while rIBV-nsp15-H238A loses the ability to inhibit protein synthesis (Fig 11D). When comparing the translation efficiency between wild-type IBV-infected and rIBV-nsp15-H238A-infected PKR knockdown cells, we find that rIBV-nsp15-H238A triggers less translation inhibition than wild-type IBV. This result further demonstrates that IBV inhibits host protein synthesis independently of the PKR-eIF2α pathway, with nsp15 playing a role in this process. When the virus loses nsp15 function, it triggers the PKR-eIF2α-mediated global translation shutdown, which is detrimental to viral protein synthesis. Previous report has demonstrated the indispensability of the 5b protein for host translation shutdown using the 5b null rIBV [55]. Xiao et al. reported that both IBV and SARS-CoV S proteins interact with eIF3F, leading to the inhibition of translation of reporter genes [58]. Screening result in S1 Fig 1 revealed that besides nsp15, expression of 5a and E also suppresses the expression of co-transfected plasmid. Combined with previous reports and the finding in this study, we conclude that IBV shuts down host protein expression by manipulating the host translation machinery through multiple viral proteins including nsp15, 5a, 5b, E, and S protein.

Subcellular localization is a critical factor influencing the proper function of proteins. Upon overexpression, nsp15 exhibits dispersed localization in both the cytoplasm and nucleus, as there is no viral RNA or viral proteins to interact and recruit nsp15 to the RTC. This distribution allows nsp15 to target cellular substrates effectively in the cytoplasm and the nucleus. During virus infection, both IBV nsp15 and MHV nsp15 [101] exhibit aggregated perinuclear localization, co-localizing with RdRp nsp12 (Fig 13B), which serves as the catalytic center of the RTC [102]. Membrane rearrangements, such as the formation of DMVs during IBV, SARS-CoV, MERS-CoV, or MHV infection [22,24,103–105], provide a microenvironment for viral RNA replication by accommodating the RTC. These membrane rearrangements are interconnected, maintaining contiguity with the endoplasmic reticulum (ER) membrane donor [24], and ribosomes are associated with the outer membrane [20,106,107]. The association of nsp15 with the RTC suggests that this EndoU may reside within ER membrane rearrangements, as indicated by the perinuclear localization of IBV nsp15 during infection and its co-localization with RTC-associated proteins like nsp12. Inside membrane rearrangements, nsp15 EndoU may cleave viral RNAs to reduce dsRNA accumulation, as demonstrated by the Dot blot and Northern blot results in this study. Recently, cryotomography revealed that DMVs of MHV contain membrane-spanning structures, including a hexameric, crown-shaped pore complex surrounding a central channel that facilitates RNA and protein transport [108]. The nucleocapsid structure was observed on the cytosolic side of the pore, suggesting that RNA is encapsulated following export from DMVs [108]. Thus, while nsp15 is associated with the RTC or membrane vesicles, it may also localize outside

membrane vesicles and target host RNA or ribosomes, thereby interfering with host mRNA translation. RIP-Seq RNA interactome analysis demonstrated that nsp15 binds to both viral RNA and cellular RNAs. Protein interactome analysis identified 809 cellular proteins interacting with nsp15. The GO and KEGG pathway enrichment analyses indicated that these nsp15-associated RNAs and proteins are involved in processes like translation, ribosomal biogenesis, mRNA splicing and processing, tRNA biosynthesis. The interaction of nsp15 with ribosomal components, supports the idea that nsp15 may directly inhibit the proper functioning of the host translation machinery. By interacting with proteins related to RNA binding, tRNA binding, dsRNA binding, RNA helicase activity, translation initiation factor activity, mRNA 5'-UTR binding, mRNA surveillance, nsp15 may directly interfere with the ribosomal translation process or disrupt RNA-binding activities. This disruption may prevent the assembly of functional translation complexes, leading to a suppression of host protein synthesis. Additionally, the enrichment of terms related to protein folding, protein stabilization, and protein quality control, including chaperone-mediated processes, suggests that nsp15 may also affect the cellular protein homeostasis. The KEGG pathway analysis revealed significant involvement of nsp15-interacting proteins in mitochondrial processes such as the TCA cycle, oxidative phosphorylation, mitochondrial translation, which are central to energy production. These pathways are critical for maintaining ATP levels required for cellular processes, including translation. Overall, nsp15's interactions with a broad range of cellular RNAs and proteins involved in RNA metabolism, translation, and protein folding highlight its multifaceted role in suppressing host protein synthesis. The specific enrichment of coronavirus disease pathway further underscores the importance of these interactions in the context of the viral life cycle and pathogenesis. In conclusion, nsp15 likely exerts its inhibitory effect on host protein synthesis through a combination of interfering with RNA processing and translation, modulating protein homeostasis, and influencing cellular energy production. These hypotheses based need to be further investigated and validated in future studies.

Inhibiting host antiviral gene expression is an important strategy for viruses to antagonize the host innate immune response. Previous studies have reported that coronaviruses employ various mechanisms to shut down host translation. α- and β-coronaviruses employ nsp1 to inhibit host protein synthesis via multiple strategies [109]: interference with ribosomal function [46,110,111], endo-nucleolytic cleavage of 5' capped non-viral mRNA leading to its degradation [46,50,52,68], interference with nucleocytoplasmic transport of host mRNA resulting in its nuclear retention [47,48], or halting translation of host mRNA by targeting mRNA derived from the nucleus [50]. Additionally, SARS-CoV-2 nsp14 and nsp16 also inhibit host translation, through a mechanism in which nsp16 suppresses global mRNA splicing and prevents the production of mature host mRNA [112,113]. In this study, by comparing the activity of wild type and catalytic-deficient nsp15, we demonstrate that besides its well-characterized activity on viral RNA, nsp15 exerts additional functions by targeting host substrates, ultimately leading to the suppression of host protein synthesis. The role of nsp15 in the regulation of host and viral protein expression is summarized in the model shown in Fig 15. Infection with IBV triggers a host translation shutoff mechanism that selectively inhibits host protein synthesis, promoting viral protein synthesis by exploiting the host's translation machinery. This process likely involves the coordinated action of nsp15 along with accessory proteins such as 5b, 5a, S, and E, as evidenced in our study (S1 Fig) and previous reports [55,58]. Nsp15 plays a crucial role in this mechanism by targeting cellular RNAs and proteins to suppress host protein synthesis. Additionally, it targets viral RNA to mitigate the accumulation of dsRNA, thus preventing the activation of PKR and subsequent phosphorylation of eIF2α, which could hinder virus replication. By regulating both cellular RNAs and viral RNAs, nsp15 aids in creating an environment conducive to efficient viral replication. The

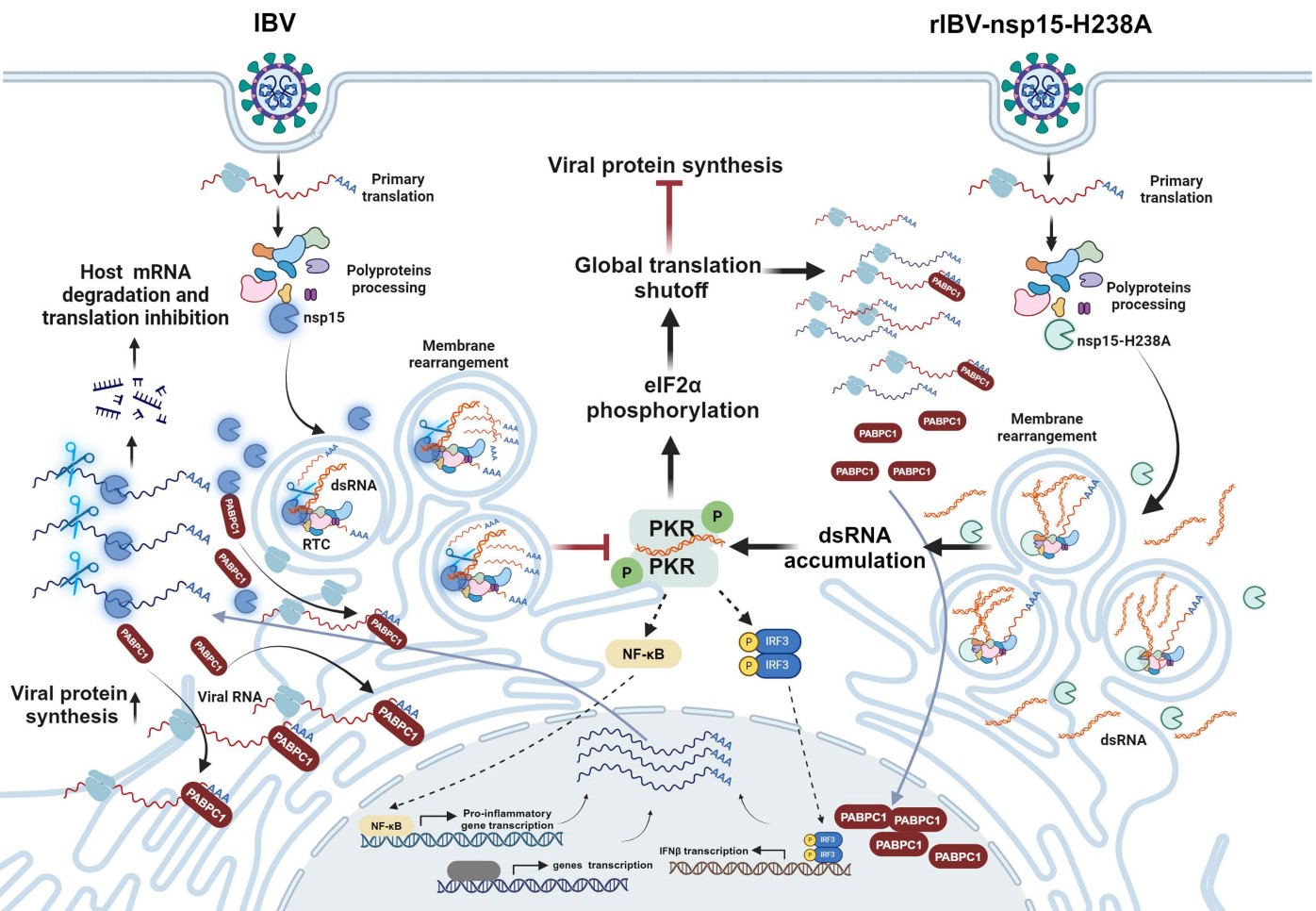

**Fig 15. Working model of the mode of action on host gene expression of nsp15 in the context of IBV infection.** Upon IBV infection, nsp15 plays a crucial role in regulating both viral and host protein synthesis simultaneously. It localizes to the RTC, where it regulates the abundance of viral dsRNA intermediates. This function is essential for the virus to evade the activation of the PKR-eIF2α pathway, which can lead to global protein synthesis inhibition for both viral and host mRNAs. This mechanism allows the virus to evade the host's defense mechanisms and maintain translation of viral mRNAs. Meanwhile, nsp15 specifically targets host RNAs including anti-viral genes, suppressing host mRNA translation while ensuring the translation of viral mRNA, ultimately promoting viral protein synthesis. Upon catalytic mutant rIBV-nsp15-H238A infection, catalytic-deficient nsp15 is no longer able to control the levels of viral dsRNA intermediates, leading to a dsRNA-PKR-eIF2α-mediated global translation shutoff that limits both, host and viral protein synthesis, thereby leading to nuclear accumulation of PABPC1. This figure was created using Biorender (BioRender, Toronto, ON, Canada).

catalytic-deficient rIBV-nsp15-H238A loses its ability to regulate the levels of viral dsRNA. Consequently, this mutant virus triggers a dsRNA-PKR-eIF2α-mediated host shutoff mechanism. As a result, both host and viral mRNA translation are impaired, leading to a reduction in protein synthesis. This dysregulation also prompts the nuclear re-localization of PABPC1, further contributing to the restriction of viral replication. The loss of nsp15's catalytic activity disrupts the delicate balance of viral RNA levels and host translation, ultimately impeding viral replication. In summary, nsp15 coordinates the regulation of viral dsRNA levels and host protein synthesis to favor viral replication and propagation: by mitigating the activation of eIF2α-dependent translation shutdown via reducing dsRNA formation and by suppressing host protein synthesis through targeting cellular RNA. This sophisticated mechanism allows the virus to evade host immune responses and maintain efficient translation of viral proteins for successful infection.

This novel finding regarding the role of nsp15 in regulating host and viral gene expression enhances our understanding of the mechanisms by which coronaviruses manipulate host translation. Given that EndoU enzymes are conserved genetic markers of *Nidovirales*, the discovery of nsp15's function as a host translation suppressor may represent a breakthrough in uncovering common strategies employed by coronaviruses and possibly other nidoviruses. This insight could pave the way for the development of broad-spectrum antiviral strategies targeting these shared mechanisms.

## Materials and Methods

### Cells and Viruses

Human non-small cell lung carcinoma H1299 cells were purchased from the Cell Bank of the Chinese Academy of Sciences (Shanghai, China). Chicken embryo fibroblasts DF-1 cells, African green monkey kidney epithelial Vero cells, and human embryonic kidney HEK293T cells were obtained from ATCC. Porcine kidney epithelial cells (PK15) were provided by Prof. Hongjun Chen (Shanghai Veterinary Research Institute, CAAS, China), while LLC-PK1 and ST cells were provided by Prof. Tonglin Shan (Shanghai Academy of Agricultural Sciences, CAAS, China). H1299 cells were cultured in Roswell Park Memorial Institute 1640 medium (RPMI, 21875034, Gibco) supplemented with 10% (v/v) fetal bovine serum (FBS, Gibco). The remaining cell lines were maintained in Dulbecco's Modified Eagle Medium (DMEM, Gibco) supplemented with 10% FBS.

A mammalian cell adapted Beaudette IBV strain obtained from Prof Dingxiang Liu (South China Agricultural University, China) [114] was used in this study, as this IBV strain can be propagated in the DF-1 cells as well as in some mammalian cell lines, including Vero and H1299 cells [115]. The recombinant virus rIBV-nsp15-H238A was retrieved, and its generation was described in detail in our previous study [116].

### Plasmids

V5-tagged constitutively active form (N-terminus 1-1920 bp) of chicken MDA5 [V5-chMDA5(N)] and full-length chicken IRF7 (V5-chIRF7) were cloned into pcDNA 3.1 and HA-tagged full-length chicken MAVS was cloned into pCAGGS [117]. The epitope tag is located at the C-terminus of the inserted gene. The plasmid pEGFP-N1 was purchased from Addgene (6085-1, USA). Plasmids encoding IBV nsp2, nsp3, nsp4, nsp5, nsp6, nsp7, nsp8, nsp9, nsp10, nsp12, nsp13, nsp14, nsp15, nsp16, 3a, 3b, 5a, 5b, S, E, M, N, IAV NS1, PEDV nsp15, TGEV nsp15, SARS-CoV-1 nsp15, SARS-CoV-2 nsp15, and the catalytic-deficient mutants of nsp15 were cloned into PXJ40F, as previously described [30]. MERS-CoV nsp15 cDNA was purchased from Sangon Biotech (China), and PDCoV-nsp15 cDNA was provided by Prof. Tonglin Shan (Shanghai Veterinary Research Institute, CAAS, China); both were inserted into PXJ40F.

The oligomerization-deficient mutants of IBV nsp15 (D285A and D315A), and the catalytic-deficient mutants of MERS-CoV nsp15 and PDCoV nsp15 (MERS-CoV nsp15-H231A and PDCoV nsp15-H219A), were cloned using the Mut Express II Fast Mutagenesis Kit V2 (C214, Vazyme). The mutagenesis primers used were: for IBV nsp15-D285A, 5'-TGTTGT<u>GGC</u>TTTACTGCTTGATGATTTCTTAGAACTTC-3' (F) and 5'-GCAG TAAA<u>GCC</u>ACAACAGTACACACTTGCTTGTAA-3' (R); for IBV nsp15-D315A, 5'-GTGTCAATT<u>GCT</u>TACCATAGCATAAATTTTATGACTTGG-3' (F) and 5'-TGGTA <u>AGC</u>AATTGACACTGTTACAACTTTTGACTT-3' (R); for MERS-CoV nsp15-H231A, 5'-TTTTGAG<u>GCC</u>GTAGTCTATGGAGACTTCTCTCATACTACG-3' (F) and 5'-AGAC TAC<u>GGC</u>CTCAAAAGCATAGTTTTCCAAGCC-3'(R); for PDCoV nsp15-H231A,

5'-CGGAACTGCCACACTTATCTCACTAGTTAAAAACAAGTTTG-3' (F) and 5'-TAAGT GTGGCAGTTCCGCCAATGACTGGACTG -3' (R). The underlined sequences indicate the targeted sites for the mutations.

## Antibodies

For Western blot analysis, mouse anti-V5 (Thermo Fisher Scientific, #R961-25, HRP-conjugated), mouse anti-HA (MBL, #M180-7, HRP-conjugated), and mouse anti-Flag (MBL, #M185-7, HRP-conjugated) antibodies were diluted to 1:2500 before use; mouse anti-β-actin (CST, #3700S), rabbit anti-GFP (CST, #2956), chicken anti-Flag (Gentaur, #AFLAG), rabbit anti-phosphorylated PKR (Abcam, #ab32036), rabbit anti-PKR (CST, #12297), rabbit anti-phosphorylated eIF2α (CST, #3398), rabbit anti-eIF2α (CST, #5324), rabbit anti-RPS6 mAb (CST, #2217), and rabbit eIF4E mAb (CST, #2067) were diluted to 1:1000 dilution before use; rabbit anti-IBV N was diluted to 1:2000 before use. For immunofluorescence, rabbit anti-human PABPC1 (Abcam, #Ab21060,), rabbit anti-IBV-N, rabbit anti-IBV nsp12 pAb (provided by Prof Dingxiang Liu, South China Agricultural University, China), and mouse anti-IBV nsp15 mAb (provided by Dr. Min Liao's lab, Zhejiang University, China) were diluted to 1:500 before use. Mouse anti-puromycin (Sigma-Aldrich, #MABE343) was diluted to 1:25000 for Western blot analysis and to 1:10000 for immunofluorescence; mouse anti-dsRNA mAb J2 (Scicons, #10010200) was diluted to 1:1000 for dot blot analysis.

Goat anti-rabbit IgG (ABclonal, #AS014, HRP-conjugated) and goat anti-mouse IgG (ABclonal, #AS003, HRP-conjugated) were diluted to 1:5000 for Western blot analysis or dot blot analysis. For immunofluorescence assay, goat anti-chicken IgY (Invitrogen, # A-11041, Alexa Fluor 568-conjugated), goat anti-mouse IgG (Invitrogen, #A-11029, Alexa Fluor 488-conjugated), and goat anti-rabbit IgG (Invitrogen, #A-11034, Alexa Fluor 488-conjugated) were diluted to 1:500 before use.

## Chemicals

Here is a list of the reagents and their respective catalog numbers or descriptions: puromycin (Merck, #58-58-2, reconstituted with sterile ddH$_2$O to 50 mg/mL as stock solutions); Fugene (HD) (Promega, #E2311); Opti-MEM (Gibco, #31985062); TRIzol regent (Life Technologies, #15596018); M-MLV (Promega, #M1701); Random primers (Invitrogen, #48190011); SYBR green master mix (Dongsheng Biotech, #P2092); 4%-20% gradient SurePAGE gel (Gen-Script, #M00657); Tris-MOPS-SDS running buffer (GenScript, #M00138); Biotin-oligo d(T) (Promega, #Z5261, 0.2 μmol/L for mRNA FISH); Diethyl Pyrocarbonate (DEPC)-treated water (Invitrogen, #4387937); Paraformaldehyde (Sigma-Aldrich, #158127); Triton X-100 (Sigma-Aldrich, #X100); BSA (Sigma-Aldrich, #A2153); SSC (Invitrogen, #15557044); PBS (Sigma-Aldrich, #P3813); Dithiothreitol (DTT) (Thermo Scientific, #R0861); RNase inhibitor (Promega, #N2611); Streptavidin (Invitrogen, # SA1001, FITC -conjugated, 1:500 dilution for FISH experiment); Diamidino-2-phenylindole (DAPI) (Thermo Scientific, #62247, 1:1000 dilution for immunofluorescence assay); Mounting medium (Sigma-Aldrich, #C9368); ARS (Sigma-Aldrich, #S7400).

## Plasmid transfection

Plasmid transfection was performed as described previously [30] using Fugene HD. Briefly, plasmid(s) and Fugene HD were mixed at a ratio of plasmid to Fugene HD of 1:3 in Opti-MEM. After incubating the mixture for 15 minutes at room temperature, it was added to the cultured cells.

### RNA interference assay

H1299 cells were seeded in six-well culture plates and transfected with PKR-specific small interfering RNA (siPKR) or negative control (siRNA NC) duplexes at a concentration of 50 nM using Lipofectamine 2000 (Invitrogen) according to the manufacturer's instructions. The sequences of the siPKR duplexes were as follows: Sense: 5′-UUGGUACAGGUUCUA CUAAUU-3′;antisense: 5′-UUAGUAGAACCUGUACCAAUU-3′. At 48 h post-transfection (h.p.i.), the cells were infected with IBV or rIBV-nsp15-H238A at a multiplicity of infection (MOI) of 1. At 18 or 24 h post-infection (h.p.i.), the cells were incubated with 5 μg/mL puromycin for 1 h at 37°C, followed by Western blot analysis.

### Dual luciferase assay

Plasmids encoding IBV proteins (300 ng each, CMV promoter), human/chicken MDA5 (200 ng each, CMV promoter), or human/chicken MAVS (200 ng each, SV40 promoter), an IFNβ promoter-driven Firefly luciferase reporter (100 ng), and a HSV TK promoter-driven Renilla luciferase reporter pRL-TK (50 ng) were co-transfected into cells in a 24-well plate. To ensure comparability between different IBV protein groups, all plasmids except those encoding IBV proteins were mixed and divided into equal volumes, to which the plasmid encoding each IBV protein was added. Each co-transfection group was repeated twice. At 24 h.p.t., cells were lysed using the passive lysis buffer supplied by the Dual-Luciferase Reporter Assay System (Promega, #E1910). Firefly luciferase activity was measured by adding LAR II, and Renilla luciferase activity was measured by adding Stop & Glo, according to the manufacturer's instructions, using a luminometer (Cytation 5 imaging multimode reader, Biotek).

### Western blot analysis

Cells transfected with plasmids for 24 h were lysed in lysis buffer, and the cell lysates were resolved on a 10% SDS-PAGE gel. To separate IBV-nsp7 (23 kDa), nsp8 (12 kDa), nsp9 (15 kDa), nsp12 (106 kDa), and nsp15 (37 kDa) on the same gel, a commercial 4-20% acrylamide gradient SurePAGE gel was used. Unlike SDS-PAGE gel electrophoresis, SurePAGE gel electrophoresis was performed using 1X Tris-MOPS-SDS running buffer. The proteins separated by the SDS-PAGE were transferred to a nitrocellulose membrane (GE Life Sciences). The membrane was then incubated in blocking buffer (5% non-fat milk powder diluted in TBST: 20 mM Tris, 150 mM NaCl, 0.1% Tween 20 detergent) for 1 h at room temperature or overnight at 4°C. After blocking, the membrane was incubated with the primary antibody diluted in blocking buffer, followed by three washes in TBST. Next, the membrane was incubated with an HRP-conjugated secondary antibody diluted in blocking buffer and washed three more times in TBST. Finally, the signal was detected using a Tanon 4600 Chemiluminescent Imaging System (Bio Tanon, China) after development with a luminol chemiluminescence reagent kit (Share-bio, China).

### Puromycin labelling

Puromycin resembles the 3′ end of tRNA and binds to growing peptide chains during translation, causing the cessation of protein synthesis and release of premature polypeptides containing puromycin [79]. Cells were transfected with plasmids for 24 h or infected with IBV. Following this, the cells were incubated with 5 μg/mL puromycin for 1 h at 37°C. Equal amounts of cells cultured in six-well plates were lysed for Western blot analysis, while cells cultured in four-well chamber slides were fixed for indirect immunofluorescence assay. The intensities of puromycin-labeled peptide signals from Western blot or immunofluorescence images were quantified using the ImageJ software (NIH, USA).

## Sodium arsenite (ARS) treatment

Cells were seeded in six well culture plates and treated with 1 mM ARS (Millipore Sigma, USA) for 30 min before being collected for *in situ* hybridization and indirect immunofluorescence analysis.

## Indirect immunofluorescence assay

Briefly, cells were fixed with 4% paraformaldehyde for 15 minutes at room temperature, permeabilized with 0.5% Triton X-100 for 15 minutes at room temperature, and incubated in blocking buffer (3% BSA diluted in PBS) for 1 h at 37°C. Cells were washed with PBS three times (5 minutes each) at the intervals of the above steps on a shaker. Cells were then incubated with the primary antibody diluted in blocking buffer for 1 h at 37°C, followed by incubation with FITC- or TRITC-conjugated secondary antibody diluted in blocking buffer for 1 h at 37°C. For double staining, cells were subsequently incubated with the other primary antibody, followed by incubation with the corresponding FITC- or TRITC-conjugated secondary antibody. At the intervals of each incubation step, cells were washed three times (5 minutes each) with PBS containing 0.2% Triton X-100 on a shaker. DAPI was then applied to stain the nuclei for 7 minutes. Finally, cells were washed three times with PBS, mounted onto glass slides using ProLong Gold Antifade Mountant (Invitrogen), and examined using a Zeiss LSM880 confocal microscope.

### mRNA fluorescence *in situ* hybridization (FISH) and indirect immunofluorescence assay

To visualize subcellular mRNA localization, cells were fixed with 4% paraformaldehyde for 15 minutes, permeabilized with 0.5% Triton X-100 for 15 minutes, and blocked with 3% BSA for 1 h, followed by endo-biotin blocking using a blocking kit (Invitrogen, #E21390) according to the manufacturer's instructions [118]. Cells were then incubated with the primary antibody for 1 h at 37°C. For double staining, the other primary antibody was incubated in the same manner. At the intervals of each incubation step, cells were washed three times with DEPC-treated PBS containing 0.2% Triton X-100. Cells were again fixed with 4% paraformaldehyde and washed three times with DEPC-treated PBS. Cells were then equilibrated in 2× SSC (1 mg/mL t-RNA, 10% dextran sulfate, and 25% formamide) for 15 minutes at 42°C, followed by hybridization of biotin-oligo d(T) with the poly(A) tail of mRNA for approximately 12 h at 42°C in a humid environment. Biotin-oligo d(T) (0.2 µmol/L) was diluted in DEPC-treated PBS containing 0.2% Triton X-100, 1 mM DTT, and 200 units/mL RNase inhibitor. After the hybridization step, samples were washed with 2× SSC for 15 minutes and then with 0.5× SSC for 15 minutes at 42°C on a shaker. Cells were again fixed with 4% paraformaldehyde and washed with DEPC-treated PBS. Cells were then incubated with Alexa Fluor-conjugated secondary antibodies for 30 minutes, followed by FITC-conjugated streptavidin for 30 minutes at 37°C. At the intervals of each step, cells were washed with DEPC-treated PBS containing 0.2% Triton X-100 three times. DAPI was then applied to stain the nuclei for 7 minutes. Cells were washed again three times and mounted onto glass slides using mounting reagent. Cells were examined using a Zeiss LSM880 confocal microscope.

## *In vitro* translation

A mixture containing 0.5 µg of PXJ40-IBV-nsp15, 0.5 µg of reporter gene (PXJ40-IBV-N, PXJ40-IBV-M, or T7 luciferase control DNA), 40 µl TnT Quick Master Mix (L1170, Promega), 1 µl Methionine (1 mM), and nuclease-free water (to a final volume of 50 µl) was gently mixed by pipetting. This mixture was then incubated for 90 minutes at 30°C. Samples of the translation reaction products were analyzed by Western blot to detect protein

expression levels or by luciferase assay to measure luciferase activity. For the luciferase assay, 2.5 μl of translation reaction products were mixed with 50 μl of Luciferase Assay Reagent (Promega) by gently pipetting and then subjected to luminometer reading using a Cytation 5 imaging multimode reader (Biotek).

## Dot blot analysis

To detect dsRNA accumulation, total cellular RNAs were extracted using Trizol reagent. Subsequently, 2 μg of RNA from each experimental group was spotted onto a Hybond-N+ membrane (GE Healthcare) and UV-cross-linked (120 mJ/cm2) using a SCIENTZ 03-II instrument (Scientz Biotech). The membrane was blocked with 5% non-fat milk dissolved in DEPC-treated TBS, then incubated with mouse anti-dsRNA J2 antibody overnight at 4°C. Following this, the membrane was incubated with HRP-conjugated goat anti-mouse secondary antibody for 1 h at room temperature. Between each incubation step, the membrane was washed three times with washing buffer (0.1% Tween 20 detergent diluted in TBS). Finally, dsRNA signals were detected using a Tanon 4600 Chemiluminescent Imaging System after development with a luminol chemiluminescence reagent kit.

## Northern blot analysis

For the IBV probe preparation, total RNA was extracted from IBV-infected Vero cells using Trizol reagent following the manufacturer's protocol. Subsequently, 1 μg of total RNA was reverse transcribed using Expand Reverse Transcriptase (TransGen, China) and oligo-dT primers. The resulting cDNAs were amplified by PCR using a DIG labelling kit (Roche, USA). The primer pairs used in the PCR were: forward primer 5′-IBV-F (5′-TTTAGCAGAACATTTTGACGCAGAT-3′) and reverse primer 3′-IBV-R (5′-TTAGTAGAACCAACAAACACGACAG-3′).

For the Northern blot analysis, 30 μg of total RNA from each sample was mixed with RNA loading buffer and denatured at 65°C for 10 minutes. The denatured total RNA samples were loaded onto a 0.7% agarose formaldehyde gel for electrophoresis. Following electrophoresis, the RNAs were transferred onto a Hybond N+ membrane (GE Healthcare, USA) by capillary transfer with 20x SSC overnight at room temperature. The transferred RNAs were fixed by UV crosslinking using a Stratalinker (Stratagene, USA). The DIG-labelled specific IBV DNA probes, denatured at 100°C for 10 minutes and immediately chilled on ice for 5 minutes, were incubated with the membranes at 42°C for 12 h. After hybridization, the membranes were washed twice with 2x SSC, 0.1% SDS at 25°C, followed by two washes with 0.1x SSC, 0.1% SDS at 68°C. The membranes were then blocked in blocking buffer for 30 minutes and then incubated with anti-DIG antibody (diluted anti-DIG-AP 1:10,000 in blocking buffer) for 2 h at room temperature. After washing the membrane twice with washing buffer, signals were detected using CDP-Star chemiluminescence substrate (Roche, USA) according to the manufacturer's instructions.

## RNA immunoprecipitation (RIP) sequencing analysis

RNA immunoprecipitation was performed using the Pure Binding RNA Immunoprecipitation Kit (GENESEED, P0101, China). In brief, H1299 cells ($1 \times 10^7$ cells) were infected with IBV at an MOI of 1. After 18 h of infection, the medium was removed, and the cells were washed twice with phosphate-buffered saline (PBS) and lysed in lysis buffer containing protease inhibitors and RNase inhibitors. H1299 cell lysates were immunoprecipitated with 5 μg of mouse IgG or mouse anti-IBV nsp15 antibody, pre-bound to protein A + G beads. The RNA was extracted and purified according to the manufacturer's instructions and analyzed via

high-throughput sequencing. Sequencing services were provided by Personal Biotechnology Co., Ltd., Shanghai, China. Each experiment was performed in triplicate.

## Protein interactome analysis

H1299 cells ($1 \times 10^7$) were infected with wild-type IBV at a MOI of 1. After 18 h of infection, the culture medium was removed, and the cells were washed twice with phosphate-buffered saline (PBS). Whole-cell lysates were prepared and subjected to immunoprecipitation using 5 µg of either mouse IgG or mouse anti-IBV nsp15 monoclonal antibody pre-bound to protein A + G beads. The samples were washed five times with wash buffer to remove non-specific binding proteins. The protein-bound beads were then collected and processed for protein identification via liquid chromatography-mass spectrometry (LC-MS) analysis, performed at Shanghai Personal Biotechnology Co., Ltd. (Shanghai, China).

## Real-time quantitative RT-PCR analysis

DF-1 cells were seeded in 6-well plates and transfected with nsp15 and the catalytic deficient mutants. At the specified time points, cells were transfected with poly I:C (20 µg/mL) or treated with IFNβ (1000 IU/mL) or TNFα (20 ng/mL). DMSO treatment was included in parallel experiments as a negative control. Total cellular RNA was extracted using Trizol reagent (Ambion, Austin, TX). cDNA was synthesized by reverse transcription using the EasyScript One-Step gDNA Removal and cDNA Synthesis SuperMix kit (Trans, AE311, China) with oligo dT primer. The cDNA served as a template for real-time qPCR using SYBR green master mix (Dongsheng Biotech, China) and corresponding primers. Real-time qPCR was conducted in the CFX-96 Bio-rad instrument (Bio-rad, USA), and the specificity of the amplified PCR products was confirmed by melting curve analysis after each reaction. The primers used for IFNβ, IFITM3, and IL-8 in this study are listed in S1 Table. Three confirmatory experiments were conducted, and the representative results were shown. Statistical analysis was performed using GraphPad Prism 8 software. The data are presented as bar graphs, with error bars representing the ± standard deviation (SD) of three technical replicates within single experiment. Significance was determined using the two-tailed independent Student's t-test ($P < 0.05$) between two groups. Statistical significance was denoted as follows: ns (not significant), $P > 0.05$; *$P < 0.05$; **$P < 0.01$; ***$P < 0.001$; ****$P < 0.0001$.

## Densitometry

The ImageJ software (NIH, USA) was utilized to quantify the intensities of bands in Western blot analyses, dot blot analyses for dsRNA, as well as the signal intensity of puromycin in immunofluorescence images. Additionally, Pearson's correlation coefficient was calculated using ImageJ to analyze signal correlations in immunofluorescence images.

## Supporting information

**S1 Fig. IBV nsp15 downregulates MAVS-mediated IFNβ induction and the expression of co-transfected plasmid encoding huMAVS.** 293T cells were seeded in 96-well plates at a density of $1.5 \times 10^4$ cells/well for the luciferase assay or in 12-well plates at a density of $1 \times 10^5$ cells/well for Western blot analysis. Cells were co-transfected with plasmids encoding Flag-tagged IBV proteins (nsp2, nsp3, nsp4, nsp5, nsp6, nsp7, nsp8, nsp9, nsp10, nsp12, nsp13, nsp14, nsp15, nsp16, 3a, 3b, 5a, 5b, S, M, N, E), along with HA-huMAVS, the reporter plasmid encoding firefly luciferase driven by the inducible IFNβ promoter, and the control plasmid pRL-TK encoding Renilla luciferase driven by the constitutive HSV TK promoter. In parallel

experiments, PXJ40 and a plasmid encoding Flag-tagged IAV NS1 were individually transfected as controls. After 24 h, cells in the 96-well plates were lysed, and the firefly and Renilla luciferase activities were measured. The activity of the IFNβ promoter was normalized to Renilla and presented relative to the PXJ40 control. The bars represent the average of two independently performed co-transfection experiments. Cells in the 12-well plates were lysed and subjected to Western blot analysis to verify protein expression. The membranes were first probed with an anti-HA antibody to detect huMAVS, followed by re-probing with an anti-Flag antibody to detect IBV proteins and IAV NS1 protein. Finally, the membranes were probed with an anti-actin antibody to detect actin as a loading control. The red arrows indicate the IBV protein bands.
(TIF)

**S2 Fig. Nsp15 from various coronaviruses inhibits de novo protein synthesis in porcine cells and Vero cells.** (A-B) Porcine cells (PK1, ST, and PK15) and Vero cells were transfected with wild-type nsp15 or the corresponding catalytic-deficient nsp15 from the indicated coronaviruses. At 23 h.p.t., cells were treated with puromycin (5 μg/mL) for 1 h. Indirect immunofluorescence staining was performed to detect nsp15 (magenta) and puromycin-labelled *de novo* synthesized peptides (green). Nuclei were labelled with DAPI (blue). Fluorescence intensity of nsp15 and puromycin signal along the white line (from a to b) is indicated by histogram plot. Representative images are shown.
(TIF)

**S3 Fig. Distribution of IBV nsp15 overlaps with that of mRNA.** (A-B) Wild-type IBV nsp15 or the catalytic/oligomerization-deficient nsp15-H223A was transfected into DF-1 or Vero cells. After 24 h, *in situ* hybridization followed by indirect immunofluorescence was performed to visualize PABPC1 (red), nsp15 (magenta), and cellular mRNA (green). Nuclei were labelled with DAPI (blue). Representative images are shown. Yellow arrows indicate cells with nuclear accumulation of PABPC1. white arrows indicate cells expressing nsp15, and red arrows indicate the distribution of mRNA in cells expressing nsp15. Representative images are shown.
(TIF)

**S1 Table. Primer sequences used for real-time qPCR.**
(XLSX)

**S1 Data. The interactome of IBV nsp15 with cellular RNAs and viral RNAs.**
(XLSX)

**S2 Data. The interactome of IBV nsp15 with host cellular proteins.**
(XLSX)

## Acknowledgments

We would like to express our gratitude to Prof. Dingxiang Liu (South China Agricultural University, China) for providing the IBV Beaudette strain and the rabbit anti-IBV-nsp12 polyclonal antibody, as well as for his invaluable scientific advice. We also extend our thanks to Wenlian Weng (Shanghai Academy of Agricultural Sciences, CAAS) for constructing the rIBV-nsp15-H238A EndoU-deficient strain. Additionally, we are thankful to Prof. Bin Li (Jiangsu Academy of Agricultural Sciences, China) for providing the TGEV nsp15 cDNA, Prof. Tonglin Shan (Shanghai Academy of Agricultural Sciences, CAAS) for providing LLC-PK1 and ST cells and pDCoV cDNA, and Dr. Min Liao (Zhejiang University, China) for providing the IBV nsp15 monoclonal antibody.

## Author contributions

**Conceptualization:** Xiaoqian Gong, Maria Forlenza, Ying Liao.

**Formal analysis:** Xiaoqian Gong, Shanhuan Feng, Maria Forlenza, Ying Liao.

**Funding acquisition:** Ying Liao.

**Investigation:** Xiaoqian Gong, Shanhuan Feng, Jiehuang Wang, Bo Gao, Wenxiang Xue, Hongyan Chu, Yanmei Yuan, Yuqiang Cheng.

**Project administration:** Maria Forlenza, Ying Liao.

**Resources:** Min Liao, Lei Tan, Cuiping Song, Xusheng Qiu.

**Supervision:** Shouguo Fang, Yingjie Sun, Chan Ding, Ying Liao.

**Writing – original draft:** Xiaoqian Gong, Edwin Tijhaar, Maria Forlenza, Ying Liao.

**Writing – review & editing:** Edwin Tijhaar, Maria Forlenza, Ying Liao.

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
