## [Decision Letter · Decision Letter 0]

7 Oct 2024

Dear Dr. Liao,

Thank you very much for submitting your manuscript "Coronavirus endoribonuclease nsp15 induces host protein synthesis shutdown and ensures viral protein synthesis via evading PKR-eIF2α-mediated anti-viral translation shutoff" for consideration at PLOS Pathogens. As with all papers reviewed by the journal, your manuscript was reviewed by members of the editorial board and by several independent reviewers. In light of the reviews (below this email), we would like to invite the resubmission of a significantly-revised version that takes into account the reviewers' comments.

Revisions should address the specific points made by each reviewer. We believe that these suggestions will help to improve your manuscript even further. 

In particular, please pay careful attention to reviewer concerns regarding the following:

Cytotoxicity analysis for nsp15 overexpressionContradictory results when comparing the impact of nsp15 overexpression on exogenous and *de novo* protein synthesis inhibitionThe role for endonuclease activity in changes in host gene expression or localization of cellular signaling molecules  Contradictory results following infection and overexpression, including differences in localization of PABPC1 and the relevance of this protein in generalThe reproducibility and significance of protein synthesis inhibition as it relates to both the role of nsp15 endonuclease activity and the role of PKR. Clarification of methods used to quantify puromycin labelingConclusions that may overstate the data

We cannot make any decision about publication until we have seen the revised manuscript and your response to the reviewers' comments. Your revised manuscript is also likely to be sent to reviewers for further evaluation.

Sincerely,

Emily Ledgerwood, PhD

Guest Editor

PLOS Pathogens

Ashley St. John

Section Editor

PLOS Pathogens

Michael Malim

Editor-in-Chief

PLOS Pathogens

orcid.org/0000-0002-7699-2064

Reviewer's Responses to Questions

**Part I - Summary**

Reviewer #1: Non-structural proteins play an important role in the life cycle of viruses. These non-structural proteins play indispensable roles to suppress the physiological activity and immunity of host cells. In this paper, they studied showed the coronavirus endoribonuclease nsp15 inhibits host protein synthesis by targeting host RNA. This study revealed that nsp15 may regulate host protein expression by targeting host and viral RNA. While this result is somewhat convincing, a few concerns need to be addressed.

Reviewer #2: This resubmitted manuscript explores the role of CoV nsp15 in the induction of translational shutoff. While the role of nsp15 in reduction of viral dsRNA and prevention of PKR activation has previously been explored (Kinlder; Deng), the authors here show that this also holds true for another CoV, IBV. New findings, such as targets of nsp15 binding (including several RNA targets that regulated RNA expression) are novel, however some of the conclusions are overstated based on the data provided.

**Part II – Major Issues: Key Experiments Required for Acceptance**

Reviewer #1: 1.Fig1 A, please explain why there are the low expression level of GFP in the absence of an inhibitor. I have another concern whether the expression of β-actin protein will be inhibited, since nsp15 inhibits host protein synthesis. This problem was common in the description that follows. Fig1 B, the authors needed to indicate whether the protein is cytotoxic, since nsp15 inhibits host protein synthesis.

2.Fig4 showed some conflict with fig3. In the fig, the groups of MERS-CoV, SARS-CoV2 showed some ability of inhibition, however, the group of PDCoV showed weak ability of inhibition in protein synthesis. There are similar problem in S2 fig.

3.Fig8, Although it has been reported in the previous studies, how to make out nsp15 inhibit nuclear translocation of IRF3 in this fig.

Reviewer #2: - A number of times the authors state that differences in host gene expression and/or altered localization of signaling molecules (e.g. STAT) support the conclusion that nsp15 is targeting these host genes for degradation which is dependent on the endoU activity of nsp15. While this is a plausible hypothesis, this data alone is insufficient to support this. Alternative explanations could be that nsp15 is targeting other host RNAs that regulate expression of these proteins. This can be addressed by tempering these statements regarding what can be concluded from the data.

- The conclusion that nsp15 is important for host translational shutoff independent of PKR is appealing but unsubstantiated by the data (specifically Fig 11D). Relative quantitation of the signal in PKR siRNA kd cells infected with WT or mutant virus show 0.74 and 0.89 relative change in translation respectively. It's unclear if this difference is biologically (or statistically) significant, and it is unclear whether this quantitation data is from a single experiment or from multiple experiments. Moreover, it appears that the relative difference in puromycin incorporation in WT vs mutant infected cells (without siRNA treatment) is different in 11D vs 11B, where 11B shows substantial reduction following infection with both viruses, but 11D shows significantly more reduction in the mutant infected cells vs WT. While biological variation in such assays may be expected, it's challenging to know how consistent and biologically significant the phenotype in 11D is.

**Part III – Minor Issues: Editorial and Data Presentation Modifications**

Reviewer #1: 4.S1 fig, In the nsp15 expression group, why β-actin expressed in high level. And why some IBV protein not detected.

5.Fig2 A, why are NSP15 not detected in NSP15 transfection group. This description of the wild-type nsp15 is inaccurate, because there are only tranfection by NSP15 plasmid, but infection by virus.

6.Is the wild-type nsp15 transfeciton or viural infection? If it’s transfeciton, why the expression level was weak?

7.Fig5 B, The nuclear size varied greatly among the groups. Authors need to unify the scales.

8.Fig6, these viral nsp15 should indicated in the fig, rather than just the virus name, which can be misleading. And it's hard to make out the difference between these groups.

9.Fig11A, the labeling of these symbols(+,-) is confusing.

Reviewer #2: n/a

PLOS authors have the option to publish the peer review history of their article (what does this mean? ). If published, this will include your full peer review and any attached files.

**Do you want your identity to be public for this peer review?** For information about this choice, including consent withdrawal, please see our Privacy Policy .

Reviewer #1: No

Reviewer #2: No
---

## [Decision Letter · Decision Letter 1]

31 Dec 2024

PPATHOGENS-D-24-01388R1

Coronavirus endoribonuclease nsp15 suppresses host protein synthesis and evades PKR-eIF2 α -mediated translation shutoff to ensure viral protein synthesis

PLOS Pathogens

Dear Dr. Liao,

Thank you for submitting your manuscript to PLOS Pathogens. After careful consideration, we feel that it has merit but does not fully meet PLOS Pathogens's publication criteria as it currently stands. Therefore, we invite you to submit a revised version of the manuscript that addresses the points raised during the review process.

Please pay particular attention to the following reviewer suggestions and give them due consideration. 

Provide evidence to confirm that H1299 cells were successfully infected with wild-type viruses.Discuss the six viral transcripts and any relevant host proteins found to interact with nsp15 and relate those findings to the potential role of nsp15 in regulating viral gene expression during the viral life cycle.Clarify if nsp15 interacted with PABPC1 and relate to the proposed model. If any of these interactions were validated, for instance by IP-WB, this should be stated.

Please submit your revised manuscript within 30 days. If you will need more time than this to complete your revisions, please reply to this message or contact the journal office at plospathogens@plos.org. Please include the following items when submitting your revised manuscript:

We look forward to receiving your revised manuscript.

Kind regards,

Emily Ledgerwood, PhD

Guest Editor

PLOS Pathogens

Ashley St. John

Section Editor

PLOS Pathogens

Sumita Bhaduri-McIntosh

Editor-in-Chief

PLOS Pathogens

orcid.org/0000-0003-2946-9497

Michael Malim

Editor-in-Chief

PLOS Pathogens

orcid.org/0000-0002-7699-2064

**Journal Requirements:**

1) We notice that your supplementary Figures are included in the manuscript file. Please remove them and upload them with the file type 'Supporting Information'. Please ensure that each Supporting Information file has a legend listed in the manuscript after the references list.

2) Please ensure that the funders and grant numbers match between the Financial Disclosure field and the Funding Information tab in your submission form. Note that the funders must be provided in the same order in both places as well. Currently, this grant number "32372999 " is missing from the Funding Information tab while this grant number "31772724" is missing from the Financial Disclosure field.

Please indicate by return email the full and correct funding information for your study and confirm the order in which funding contributions should appear. Please be sure to indicate whether the funders played any role in the study design, data collection and analysis, decision to publish, or preparation of the manuscript.

**Reviewers' Comments:**

Reviewer's Responses to Questions

**Part I - Summary**

Reviewer #1: In addition to answering the reviewers' concerns, the author also supplemented RIP-Seq RNA data and GO and KEGG analyses. The paper contains a wealth of data.

Reviewer #2: This resubmitted manuscript explores the role of nsp15 in translational regulation. The authors have made significant experimental efforts to address some of the weaknesses in the originally submitted manuscript, including nsp15 interactome data.

**Part II – Major Issues: Key Experiments Required for Acceptance**

Reviewer #1: In the newly added RIP-Seq RNA data, the author mentioned infecting H1299 cells with wild-type viruses. Previous reports have shown that only the Beaudette strain of coronavirus infectious bronchitis virus after long-term domestication can infecte H1299 cell. Does the author have any evidence to confirm that the virus has successfully established an infection?

Reviewer #2: The inclusion of interactome data (RNA and protein is very interesting and provides additional support for some of the conclusions), however specific targets that were identified were not explicitly stated, only pathways and GO terms were used. This data would be even more convincing (and of interest to readers) if some discussion of specific targets were included and not simply relegated to the supplementary data file. For example, what were the 6 viral transcripts that were found to interact with nsp15? What does this potentially imply about the role of nsp15 in regulating expression of viral transcripts throughout the replication cycle? Regarding host protein interactions, was PABPC1 found to interact with nsp15? Were there other specific host proteins that were shown to interact with nsp15 that would further support the model proposed? Were some of these interactions validated using IP-WB?

**Part III – Minor Issues: Editorial and Data Presentation Modifications**

Reviewer #1: (No Response)

Reviewer #2: n/a

PLOS authors have the option to publish the peer review history of their article (what does this mean? ). If published, this will include your full peer review and any attached files.

**Do you want your identity to be public for this peer review?** For information about this choice, including consent withdrawal, please see our Privacy Policy .

Reviewer #1: **Yes: ** wenke ruan

Reviewer #2: No

**Figure resubmission:**
---

## [Editor Report · Decision Letter 2]

16 Feb 2025

Dear Dr. Liao,

We are pleased to inform you that your manuscript 'Coronavirus endoribonuclease nsp15 suppresses host protein synthesis and evades PKR-eIF2 α -mediated translation shutoff to ensure viral protein synthesis' has been provisionally accepted for publication in PLOS Pathogens.

Best regards,

Emily Ledgerwood, PhD

Guest Editor

PLOS Pathogens

Ashley St. John

Section Editor

PLOS Pathogens

Sumita Bhaduri-McIntosh

Editor-in-Chief

PLOS Pathogens

orcid.org/0000-0003-2946-9497

Michael Malim

Editor-in-Chief

PLOS Pathogens

orcid.org/0000-0002-7699-2064
---

## [Editor Report · Acceptance letter]

Dear Prof Liao,

We are delighted to inform you that your manuscript, "Coronavirus endoribonuclease nsp15 suppresses host protein synthesis and evades PKR-eIF2 α -mediated translation shutoff to ensure viral protein synthesis," has been formally accepted for publication in PLOS Pathogens.

Best regards,

Sumita Bhaduri-McIntosh

Editor-in-Chief

PLOS Pathogens

orcid.org/0000-0003-2946-9497

Michael Malim

Editor-in-Chief

PLOS Pathogens

orcid.org/0000-0002-7699-2064